# A generic construction for revocable identity-based encryption with subset difference methods

Kwangsu Lee[ID]*

Department of Computer and Information Security, Sejong University, Seoul, Korea

* kwangsu@sejong.ac.kr

**Data Availability Statement:** All relevant data are within the manuscript.

**Funding:** Kwangsu Lee was supported as part of Military Crypto Research Center (UD170109ED) funded by Defense Acquisition Program Administration (DAPA) and Agency for Defense

## Abstract

To deal with dynamically changing user's credentials in identity-based encryption (IBE), providing an efficient key revocation method is a very important issue. Recently, Ma and Lin proposed a generic method of designing a revocable IBE (RIBE) scheme that uses the complete subtree (CS) method by combining IBE and hierarchical IBE (HIBE) schemes. In this paper, we propose a new generic method for designing an RIBE scheme that uses the subset difference (SD) method instead of using the CS method. In order to use the SD method, we generically design an RIBE scheme by combining IBE, identity-based revocation (IBR), and two-level HIBE schemes. If the underlying IBE, IBR, and HIBE schemes are adaptively (or selectively) secure, then our RIBE scheme is also adaptively (or selectively) secure. In addition, we show that the layered SD (LSD) method can be applied to our RIBE scheme and a chosen-ciphertext secure RIBE scheme also can be designed generically.

## 1 Introduction

Identity-based encryption (IBE) is a new type of public-key encryption (PKE) that solve the public-key management problem in PKE by using a user's identity as a public key [1]. Since the first IBE scheme in bilinear maps was proposed by Boneh and Franklin [2], research on new types of cryptographic encryption such as IBE, hierarchical IBE (HIBE), attribute-based encryption (ABE), and predicate encryption (PE) has been actively studied as an important research topic [2–5]. Despite the long history of research on IBE, the IBE schemes have not been widely deployed in real environments. One reason of this problem is that unlike PKE schemes, which uses a public-key infrastructure to handle certificate issuance and revocation, it is not simple to revoke the private key of a user in IBE. Therefore, an important additional feature of IBE schemes is to support the private key revocation flexibly and efficiently.

The method of revoking the private key of a user in IBE has been studied since the initial IBE scheme was designed, but this method is not suitable for handling a large number of users [2]. The first revocable IBE (RIBE) scheme to efficiently handle large numbers of users was proposed by Boldyreva et al. [6]. The key design principle of their RIBE scheme is that a trusted center periodically creates and broadcasts an update key on time $T$ for non-revoked users,

Development (ADD). There was no additional external funding received for this study.

**Competing interests:** We have no conflicts of interest to disclose.

along with the generation of a user's private key. In this case, if the private key of a user *ID* is not revoked in the update key on time *T*, the user can decrypt a ciphertext for his identity *ID* and the corresponding time *T*. In other words, the RIBE scheme proposed by Boldyreva et al. can be seen as a method to support the indirect private key revocation, in which the center decides the revocation of private keys instead of the sender. Specifically, Boldyreva et al. designed their RIBE scheme by combining a tree-based broadcast method with a fuzzy identity-based encryption scheme. After the work of Boldyreva et al., various RIBE schemes and extension schemes have been proposed to enhance the efficiency, security, and functionality of RIBE [7–17].

Currently, to design an RIBE scheme, we redesign an RIBE scheme from the beginning by directly modifying an efficient IBE scheme proposed before. This is problematic in that a new RIBE scheme must be designed again whenever a new IBE scheme having a different mathematical structure is proposed. Ma and Lin recently overcome this problem by suggesting a generic method of designing an RIBE scheme by using an IBE scheme as a black-box [18]. In their generic RIBE scheme with the complete subtree (CS) method, an update key consists of $O\left(r \log \frac{N}{r}\right)$ IBE private keys and a ciphertext consists of $O(\log N)$ IBE ciphertexts where $r$ is the number of revoked users and $N$ is the number of users. In the RIBE scheme, reducing the size of update keys is an important issues since an update key should be broadcasted to all users for each time period. The motivation of this work is to reduce the update key size of the generic RIBE scheme. In tree-based broadcast encryption, there exists the subset difference (SD) method proposed by Naor et al. [19] which is more efficient than the CS method. Additionally, the layered SD (LSD) method which improved the SD method has also been proposed [20]. Therefore, we ask whether it is possible to design an RIBE scheme from an IBE scheme in a generic way using the SD/LSD method to reduce the size of update keys. If the SD/LSD method can be applied to a generic RIBE scheme, the size of an update key can be reduced from $O\left(r \log \frac{N}{r}\right)$ key elements to $O(r)$ key elements.

### 1.1 Our contributions

In this paper, we show that it is possible to design an RIBE scheme with the SD method in a generic way. As described above, the generic RIBE scheme with the CS method uses IBE and two-level HIBE schemes as basic building blocks [18]. On the contrary, our generic RIBE scheme with the SD method uses IBE, identity-based revocation (IBR), and two-level HIBE schemes as basic building blocks. The IBR scheme is a special type of identity-based broadcast encryption (IBBE) scheme in which a set *R* of revoked users is specified in an IBR ciphertext whereas a set *S* of receivers is specified in an IBBE ciphertext. The newly derived RIBE scheme with the SD method consists of $O(r)$ number of IBE and IBR private keys in an update key and $O(\log^2 N)$ number of IBE and IBR ciphertexts in a ciphertext. Compared with the previous generic RIBE scheme with the CS method, the size of an update key is reduced but the size of a ciphertext is increased. The detailed comparison of RIBE schemes is given in Table 1.

To analyze the security of our generic RIBE scheme with the SD method, we show that if the underlying IBE, IBR, and HIBE schemes are adaptively (or selectively) secure under chosen plaintext attacks, then the proposed generic RIBE scheme is also adaptively (or selectively) secure under chosen plaintext attacks. The key idea of our proof is to first divide the types of an attacker according to the queries of the attacker, and to isolate the attacker of a specific type to break the security of the underlying IBE, IBR, or HIBE scheme. However, this idea is not simple to apply since the SD method has a complicated subset cover structure unlike the CS method. To handle this complicated structure in a ciphertext, we introduce additional hybrid

**Table 1. Comparison of revocable identity-based encryption schemes.**

| Scheme | PP Size | SK Size | UK Size | CT Size | Model | DKER | Generic |
|---|---|---|---|---|---|---|---|
| BF [2] | $O(1)$ | $O(1)$ | $O(N-r)$ | $O(1)$ | SE | No | No |
| BGK [6] | $O(1)$ | $O(\log N)$ | $O(r\log\frac{N}{r})$ | $O(1)$ | SE | No | No |
| LV [7] | $O(\lambda)$ | $O(\log N)$ | $O(r\log\frac{N}{r})$ | $O(1)$ | AD | No | No |
| SE [9] | $O(\lambda)$ | $O(\log N)$ | $O(r\log\frac{N}{r})$ | $O(1)$ | AD | Yes | No |
| LLP [15] | $O(1)$ | $O(\log^2 N)$ | $O(r)$ | $O(1)$ | AD | Yes | No |
| ML1 [18] | $O(1)$ | $O(1)$ | $O(r\log\frac{N}{r})$ | $O(\log N)$ | AD | Yes | Yes |
| ML2 [18] | $O(\log N)$ | $O(1)$ | $O(r\log\frac{N}{r})$ | $O(1)$ | AD | Yes | Yes |
| Ours (SD) | $O(1)$ | $O(1)$ | $O(r)$ | $O(\log^2 N)$ | AD | Yes | Yes |
| Ours (LSD) | $O(1)$ | $O(1)$ | $O(r)$ | $O(\log^{1.5} N)$ | AD | Yes | Yes |

Let $\lambda$ be a security parameter, $N$ be the number of maximum users, and $r$ be the number of revoked users. We count the number of group elements to measure the size of parameters. We use symbols SE for selective IND-CPA and AD for adaptive IND-CPA.

games in the security proof and handle each ciphertext element of the challenge ciphertext one by one.

In addition, we show that it is possible to reduce the size of a ciphertext by extending our generic RIBE scheme to use the more efficient LSD method instead of using the SD method, but this modified scheme increases the size of an update key slightly. We also show that our generic RIBE scheme which provides only chosen-plaintext attack (CPA) security can be extended to provide the security against the more powerful chosen-ciphertext attacker (CCA). To provide the CCA security of RIBE, the underlying IBE, IBR, and HIBE schemes should provide the CCA security and a one-time signature scheme with strong unforgeability should be used.

## 1.2 Related work

**Certificate revocation.** The study of certificate revocation in public-key encryption has been the subject of much research. In reality, the most widely used certificate revocation method is to periodically issue a certificate revocation list (CRL) containing serial numbers of revoked user's certificates. In addition, a delta-CRL can be used to more efficiently issue the revocation information, and it is also possible to immediately check the state of a certificate by using the online certificate status protocol (OCSP) service. In the theoretical aspect, various certificate revocation methods which are more efficient than the traditional methods also have been proposed [21–23].

**Broadcast encryption.** Public-key broadcast encryption (PKBE) provides the revocation of receivers because a sender can specify a receiver set $S$ in a ciphertext directly [24]. Identity-based broadcast encryption (IBBE) can provide more powerful revocation than existing PKBE because the maximum number of users in the system can be exponential [25]. Identity-based revocation (IBR) can be viewed as a cryptographic scheme that implements direct user revocation because all system users except the revoked users can decrypt a ciphertext where a revoked set $R$ is specified in the ciphertext [26, 27]. However, PKBE, IBBE, and IBR have the disadvantage that a user cannot be revoked after the creation of a ciphertext. Particularly, it is a critical problem in a cryptographic system in which ciphertexts are stored in cloud storage and a user accesses these ciphertexts later since the user cannot be revoked when his or her credential is expired.

**Revocable IBE.** Boneh and Franklin [2] proposed a revocation method for IBE such that a trusted center periodically issues a private key for a user by combining an identity and time as $ID\|T$, but this method is not scalable since a secure channel should be established for every time. The efficient and scalable RIBE scheme was proposed by Boldyreva et al. [6] by combining the complete subtree (CS) method and a fuzzy identity-based encryption scheme. In their RIBE scheme, a ciphertext is associated with a receiver's identity $D$ and time $T$, and a trusted center periodically issues an update key one time $T$ for non-revoked users to implement the indirect key revocation. A number of secure and efficient RIBE schemes using a broadcast method for key updates have been proposed [7–9, 13, 14, 17, 28]. Most of the RIBE schemes follow the CS method for update keys, but Lee et al. [15] showed that an RIBE scheme with the SD method can be designed to reduces the size of update keys. Recently, Ma and Lin [18] proposed a generic RIBE construction with the CS method by combining IBE and HIBE schemes.

**Revocable HIBE.** The first revocable HIBE (RHIBE) scheme, which provides the private key revocation in HIBE, was proposed by Seo and Emura [10]. They proposed an RHIBE scheme by applying the design principle of previous RIBE schemes to an HIBE scheme. To improve the initially proposed RHIBE scheme, Seo and Emura later introduced a history-free update method to reduce the size of private keys and update keys [12]. After that, Lee and Park have introduced a new RHIBE scheme with short private keys and short updated keys by introducing an intermediate private key in HIBE and using a modular design method [16]. In order to enhance the selective security of previous RHIBE schemes, Lee [29] proposed an adaptively secure RHIBE scheme by applying the dual system encryption method of Waters [30], which was successfully used in adaptively secure IBE and HIBE schemes.

**Revocable ABE.** ABE is an extension of IBE in which a ciphertext is associated with attributes and a private key is associated with an access structure, and the ciphertext of ABE can be decrypted by the private key of ABE if the attributes satisfies the access structure [4]. An revocable ABE (RABE) scheme was proposed by Boldyreva et al. [6] by following the design principle of their RIBE scheme. ABE is well-suited for environments such as cloud storage where multiple users access different ciphertexts since it can provide flexible access control. For such an environment, Sahai et al. [31] proposed a revocable-storage ABE (RS-ABE) scheme that supports ciphertext updates as well as user key revocation. Lee et al. [29, 32] proposed an improved RS-ABE scheme by using a self-updatable encryption scheme, and they also proposed an RS-ABE scheme that provides the CCA security [33]. A generic construction of ABE with direct revocation in which a revoked set is attached in a ciphertext was proposed by Yamada et al. [34].

## 2 Preliminaries

In this section, we first review the definition and security model of IBE, IBR, and HIBE. Next, we review the definition and security model of RIBE.

### 2.1 Identity-based encryption

Identity-based encryption (IBE) is a kind of public key encryption (PKE) that can use a receiver's identity as a public key [2]. In IBE, a sender generates a ciphertext by encryption a message for the receiver's identity ID. A receiver retrieves a private key corresponding to his identity $ID$ from a trusted center and then decrypts the ciphertext if the identity of the ciphertext is equal to the identity of the private key. The detailed syntax of IBE is given as follows.

**Definition 2.1** (Identity-Based Encryption, IBE). *Let $\mathcal{I}$ be an identity space and $\mathcal{M}$ be a message space. An IBE scheme consists of four algorithms **Setup**, **GenKey**, **Encrypt**, and **Decrypt**, which are defined as follows*:

***Setup***($1^\lambda$): *The setup algorithm takes as input a security parameter $1^\lambda$. It outputs a master key MK and public parameters PP.*

***GenKey***(*ID, MK, PP*): *The private key generation algorithm takes as input an identity $ID \in \mathcal{I}$, the master key MK, and public parameters PP. It outputs a private key $SK_{ID}$.*

***Encrypt***(*ID, M, PP*): *The encryption algorithm takes as input an identity $ID \in \mathcal{I}$, a message $M \in \mathcal{M}$, and public parameters PP. It outputs a ciphertext $CT_{ID}$.*

***Decrypt***($CT_{ID}$, $SK_{ID'}$, *PP*): *The decryption algorithm takes as input a ciphertext $CT_{ID}$, a private key $SK_{ID'}$, and public parameters PP. It outputs a message M.*

*The correctness of IBE is defined as follows: For all MK and PP generated by **Setup**($1^\lambda$), $SK_{ID}$ generated by **GenKey**(ID, MK, PP) for any ID, and any ID and M, it is required that*

- ***Decrypt***(***Encrypt***(*ID, M, PP*), $SK_{ID}$, *PP*) = *M*.

The security model of IBE is defined by extending the IND-CPA security model of PKE to allow additional private key queries [2]. In this model, an attacker can request a private key of an identity *ID*. In the challenge stage, the attacker submits a challenge identity $ID^*$ and challenge messages $M_0^*, M_1^*$, and then receives a challenge ciphertext $CT^*$. The attacker further queries private keys and finally guesses the message hidden in $CT^*$. The detailed description of the security model of IBE is given as follows.

**Definition 2.2** (IND-CPA Security). *The IND-CPA security of IBE is defined in terms of the following game between a challenger $\mathcal{C}$ and a PPT adversary $\mathcal{A}$:*

1. ***Setup***: *$\mathcal{C}$ generates a master key MK and public parameters PP by running **Setup**($1^\lambda$). It keeps MK to itself and gives PP to $\mathcal{A}$.*

2. ***Phase 1***: *$\mathcal{A}$ may adaptively request private keys for identities $ID_1, \ldots, ID_{q_1}$. In response, $\mathcal{C}$ gives the corresponding private keys $SK_{ID_1}, \ldots, SK_{ID_{q_1}}$ to $\mathcal{A}$ by running **GenKey**($ID_i$, MK, PP).*

3. ***Challenge***: *$\mathcal{A}$ submits a challenge identity $ID^*$ and two messages $M_0^*, M_1^*$ with the equal length subject to the restriction: for all $ID_i$ of private key queries, $ID_i \neq ID^*$. $\mathcal{C}$ flips a random coin $\mu \in \{0, 1\}$ and gives the challenge ciphertext $CT^*$ to $\mathcal{A}$ by running **Encrypt** $(ID^*, M_\mu^*, PP)$.*

4. ***Phase 2***: *$\mathcal{A}$ may continue to request private keys for $ID_{q1}+1, \ldots, ID_q$.*

5. ***Guess***: *$\mathcal{A}$ outputs a guess $\mu' \in \{0, 1\}$ of $\mu$, and wins the game if $\mu = \mu'$.*

*The advantage of $\mathcal{A}$ is defined as $Adv_{\mathcal{A}}^{IBE}(\lambda) = |\Pr[\mu = \mu'] - \frac{1}{2}|$ where the probability is taken over all the randomness of the game. An IBE scheme is IND-CPA secure if for all PPT adversary $\mathcal{A}$, the advantage of $\mathcal{A}$ is negligible in the security parameter $\lambda$.*

## 2.2 Identity-based revocation

Identity-based revocation (IBR) is a kind of public-key broadcast encryption (PKBE) [26], in which a large number of users with identities can participate to the system and a sender can specify the set *R* of revoked users in a ciphertext instead of the set *S* of receivers. In IBR, a sender generates a ciphertext *CT* by using a revoked set *R* and a message *M*, and then broadcasts the ciphertext. A receiver retrieves a private key for his or her identity from a trusted central and decrypt the ciphertext if his or her identity is not included in the set *R*. The detailed syntax of IBR is given as follows.

**Definition 2.3** (Identity-Based Revocation, IBR). *An IBR scheme consists of four algorithms **Setup**, **GenKey**, **Encrypt**, and **Decrypt**, which are defined as follows:*

**Setup**($1^\lambda$): *The setup algorithm takes as input a security parameter $1^\lambda$. It outputs a master key MK and public parameters PP.*

**GenKey**(*ID*, *MK*, *PP*): *The private key generation algorithm takes as input an identity $ID \in \mathcal{I}$, the master key MK, and public parameters PP. It outputs a private key $SK_{ID}$.*

**Encrypt**(*R*, *M*, *PP*): *The encryption algorithm takes as input revoked identities $R \subset \mathcal{I}$, a message $M \in \mathcal{M}$, and public parameters PP. It outputs a ciphertext $CT_R$.*

**Decrypt**($CT_R$, $SK_{ID}$, *PP*): *The decryption algorithm takes as input a ciphertext $CT_R$, a private key $SK_{ID}$, and public parameters PP. It outputs a message M.*

*The correctness of IBR is defined as follows: For all MK and PP generated by **Setup**($1^\lambda$), $SK_{ID}$ generated by **GenKey**(ID, MK, PP) for any ID, and any R such that $ID \notin R$ and any M, it is required that*

- **Decrypt**(**Encrypt**(*R*, *M*, *PP*), $SK_{ID}$, *PP*) = M.

The security model of IBR is defined by extending the IND-CPA security model of PKBE to account for the revoked set R [26]. In this model, an attacker requests private key queries on identities. In the challenge step, the attacker submits a challenge revoked set $R^*$ and the challenge message $M_0^*, M_1^*$ and receives a challenge ciphertext $CT^*$. The attacker additionally requests private key queries and finally guesses the hidden message in $CT^*$. In this game, all identities of private keys must belong to the revoked set $R^*$. The detailed description of the security model is given as follows.

**Definition 2.4** (IND-CPA Security). *The IND-CPA security of IBR is defined in terms of the following game between a challenger $\mathcal{C}$ and a PPT adversary $\mathcal{A}$:*

1. **Setup**: *$\mathcal{C}$ generates a master key MK and public parameters PP by running **Setup**($1^\lambda$). It keeps MK to itself and gives PP to $\mathcal{A}$.*

2. **Phase 1**: *$\mathcal{A}$ may adaptively request private keys for identities $ID_1, \ldots, ID_{q_1}$. In response, $\mathcal{C}$ gives the corresponding private keys $SK_{ID_1}, \ldots, SK_{ID_{q_1}}$ to $\mathcal{A}$ by running **GenKey**($ID_i$, MK, PP).*

3. **Challenge**: *$\mathcal{A}$ submits a challenge revoked set $R^*$ of users and two messages $M_0^*, M_1^*$ with the equal length subject to the restriction: for all $ID_i$ of private key queries, $ID_i \in R^*$. $\mathcal{C}$ flips a random coin $\mu \in \{0, 1\}$ and gives the challenge ciphertext $CT^*$ to $\mathcal{A}$ by running **Encrypt** ($R^*, M_\mu^*, PP$).*

4. **Phase 2**: *$\mathcal{A}$ may continue to request private keys for $ID_{q1}+1, \ldots, ID_q$.*

5. **Guess**: *$\mathcal{A}$ outputs a guess $\mu' \in \{0, 1\}$ of $\mu$, and wins the game if $\mu = \mu'$.*

*The advantage of $\mathcal{A}$ is defined as $Adv_{\mathcal{A}}^{IBR}(\lambda) = |\Pr[\mu = \mu'] - \frac{1}{2}|$ where the probability is taken over all the randomness of the game. An IBR scheme is IND-CPA secure if for all PPT adversary $\mathcal{A}$, the advantage of $\mathcal{A}$ is negligible in the security parameter $\lambda$.*

## 2.3 Hierarchical identity-based encryption

Hierarchical identity-based encryption (HIBE) is an extension of IBE in which a hierarchical identity is used to represent a user's identity and the delegation of private keys is provided [3, 35]. In HIBE, a user receives a private key for his hierarchical identity from a trusted center, or receives a delegated private key from another user. If a sender creates a ciphertext for a receiver's hierarchical identity and transmits it to a receiver, then the receiver can decrypt the

ciphertext by using his private key if the hierarchical identity of his private key is a prefix of the hierarchical identity of the ciphertext.

Let $HID = (ID_1, \ldots, ID_k)$ be an identity vector of size $k$. We let $HID|_j$ be a vector $(ID_1, \ldots, ID_j)$ of size $j$ derived from $HID$. We define a function $Prefix(HID|_k)$ that returns a set of prefix vectors $\{HID|_j\}_{1 \leq j \leq k}$ where $HID|_k = (ID_1, \ldots, ID_k)$. The detailed syntax of HIBE is given as follows.

**Definition 2.5** (Hierarchical Identity-Based Encryption, HIBE). *An HIBE scheme consists of five algorithms* **Setup**, **GenKey**, **Delegate**, **Encrypt**, *and* **Decrypt**, *which are defined as follows*:

**Setup**$(1^\lambda, L_{max})$. *The setup algorithm takes as input a security parameter $1^\lambda$ and maximum hierarchical depth $L_{max}$. It outputs a master key MK and public parameters PP.*

**GenKey**$(HID|_k, MK, PP)$. *The key generation algorithm takes as input a hierarchical identity $HID|_k = (ID_1, \ldots, ID_k)[0] \in \mathcal{I}^k$ where $k \leq L_{max}$, the master key MK, and the public parameters PP. It outputs a private key $SK_{HID|_k}$.*

**Delegate**$(HID|_k, SK_{HID|_{k-1}}, PP)$. *The delegation algorithm takes as input a hierarchical identity $HID|_k$, a private key $SK_{HID|_{k-1}}$ for $HID|_{k-1}$, and the public parameters PP. It outputs a delegated private key $SK_{HID|_k}$.*

**Encrypt**$(HID|_\ell, M, PP)$. *The encryption algorithm takes as input a hierarchical identity $HID|_\ell = (ID_1, \ldots, ID_\ell) \in \mathcal{I}^\ell$ where $\ell \leq L_{max}$, a message M, and public parameters PP. It outputs a ciphertext $CT_{HID|_\ell}$.*

**Decrypt**$(CT_{HID|_\ell}, SK_{HID|_k}, PP)$. *The decryption algorithm takes as input a ciphertext $CT_{HID|_\ell}$, a private key $SK_{HID|_k}$, and public parameters PP. It outputs a message M.*

*The correctness of HIBE is defined as follows: For all MK, PP generated by* **Setup**$(1^\lambda, L_{max})$, *all $HID|_\ell, HID|_k$, any $SK_{HID|_k}$ generated by* **GenKey**$(HID|_k, MK, PP)$ *such that $HID|_k \in Prefix(HID|_\ell)$, it is required that*

- $Decrypt(Encrypt(HID|_\ell, M, PP), SK_{HID|_k}, PP) = M.$

The security model of HIBE is defined by extending the security model of IBE to include additional private key delegations [3, 35]. That is, an attacker can request delegated private key queries together with general private key queries. In this case, if the distribution of general private keys and the distribution of delegate private keys are the same, then we can only consider general private key queries to simplify the security model. The detailed security model of HIBE is given as follows.

**Definition 2.6** (IND-CPA Security). *The IND-CPA security of HIBE is defined in terms of the following game between a challenger $\mathcal{C}$ and a PPT adversary $\mathcal{A}$:*

1. **Setup**: *$\mathcal{C}$ generates a master key MK and public parameters PP by running* **Setup**$(1^\lambda, L_{max})$. *It keeps MK to itself and gives PP to $\mathcal{A}$.*

2. **Phase 1**: *$\mathcal{A}$ may adaptively request a polynomial number of private key queries. In response, $\mathcal{C}$ gives a corresponding private key $SK_{HID|_k}$ to $\mathcal{A}$ by running* **GenKey**$(HID|_k, MK, PP)$ *for each query.*

3. **Challenge**: *$\mathcal{A}$ submits a challenge hierarchical identity $HID^*|_\ell$ and two messages $M_0^*, M_1^*$ with the equal length subject to the restriction: for each $HID|_k$ of private key queries, $HID|_k \notin Prefix(HID^*|_\ell)$. $\mathcal{C}$ flips a random coin $\mu \in \{0, 1\}$ and gives a challenge ciphertext $CT^*$ to $\mathcal{A}$ by running* **Encrypt** $(HID^*|_\ell, [0]M_\mu^*, PP)$.

4. **Phase 2**: $\mathcal{A}$ may continue to request private key queries.

5. **Guess**: $\mathcal{A}$ outputs a guess $\mu' \in \{0, 1\}$ of $\mu$, and wins the game if $\mu = \mu'$.

The advantage of $\mathcal{A}$ is defined as $Adv_{\mathcal{A}}^{HIBE}(\lambda) = |\Pr[\mu = \mu'] - \frac{1}{2}|$ where the probability is taken over all the randomness of the game. An HIBE scheme is IND-CPA secure if for all PPT adversary $\mathcal{A}$, the advantage of $\mathcal{A}$ is negligible in the security parameter $\lambda$.

## 2.4 Revocable identity-based encryption

Revocable identity-based encryption (RIBE) is an extension of existing identity-based encryption (IBE) to support private key revocation [6]. In RIBE, each user receives a private key for his or her identity *ID* from a trusted center. The trusted center then periodically generates an update key which is associated with time *T* and a non-revoked user set, and then it broadcasts the update key through the public channel. In this case, if the private key of a user is not revoked in the update key, the user can derive a decryption key for *ID* and *T* by combining the private key and the update key, and this decryption key can be used to decrypt a ciphertext which is related with *ID* and *T*. The syntax of RIBE is given as follows.

**Definition 2.7** (Revocable IBE, RIBE). *An RIBE scheme consists of seven algorithms* **Setup**, **GenKey**, **UpdateKey**, **DeriveKey**, **Encrypt**, **Decrypt**, *and* **Revoke**, *which are defined as follows*:

**Setup**$(1^{\lambda})$: *The setup algorithm takes as input a security parameter $1^{\lambda}$. It outputs a master key MK, an (empty) revocation list RL, and public parameters PP.*

**GenKey**(*ID*, *MK*, *PP*): *The private key generation algorithm takes as input an identity $ID \in \mathcal{I}$, the master key MK, and public parameters PP. It outputs a private key $SK_{ID}$.*

**UpdateKey**(*T*, *RL*, *MK*, *PP*): *The update key generation algorithm takes as input update time $T \in \mathcal{T}$, the revocation list RL, the master key MK, and public parameters PP. It outputs an update key $UK_T$.*

**DeriveKey**($SK_{ID}$, $UK_T$, *PP*): *The decryption key derivation algorithm takes as input a private key $SK_{ID}$, an update key $UK_T$, and public parameters PP. It outputs a decryption key $DK_{ID,T}$.*

**Encrypt**(*ID*, *T*, *M*, *PP*): *The encryption algorithm takes as input an identity $ID \in \mathcal{I}$, time T, a message $M \in \mathcal{M}$, and public parameters PP. It outputs a ciphertext $CT_{ID,T}$.*

**Decrypt**($CT_{ID,T}$, $DK_{ID',\,T'}$, *PP*): *The decryption algorithm takes as input a ciphertext $CT_{ID,T}$, a decryption key $DK_{ID',\,T}$, and public parameters PP. It outputs a message M.*

**Revoke**(*ID*, *T*, *RL*): *The revocation algorithm takes as input an identity ID to be revoked and revocation time T, and a revocation list RL. It outputs an updated revocation list RL.*

The correctness of RIBE is defined as follows: For all MK, RL, and PP generated by **Setup**$(1^{\lambda})$, $SK_{ID}$ generated by **GenKey**(ID, MK, PP) for any ID, $UK_T$ generated by **UpdateKey**(T, RL, MK, PP) for any T and RL such that $(ID, T_j) \notin RL$ for all $T_j \leq T$, $CT_{ID,T}$ generated by **Encrypt**(ID, T, M, PP) for any ID, T, and M, it is required that

• **Decrypt**($CT_{ID,T}$, **DeriveKey**($SK_{ID}$, $UK_T$, PP), PP) = M.

The security model of RIBE was first defined by Boldyreva et al. [6], and then this security model was extended by Seo and Emura [9] to support decryption key exposure resistance. In the security model of RIBE, an attacker can request a private key query for an identity *ID*, an update key query for time *T*, a decryption key query for *ID* and *T*, and a revocation query. In the challenge step, the attacker submits a challenge identity $ID^*$, challenge time $T^*$, and challenge messages $M_0^*, M_1^*$, and receives a challenge ciphertext $CT^*$. Note that the private key

query for $ID^*$ is not allowed in the IBE security model, but this private key query for $ID^*$ is allowed in the RIBE security model. At this time, if the private key for $ID^*$ is queried, then the private key for $ID^*$ must be revoked in the update key on the challenge time $T^*$. The detailed definition of the RIBE security model is given as follows.

**Definition 2.8** (IND-CPA Security). *The IND-CPA security of RIBE is defined in terms of the following experiment between a challenger $\mathcal{C}$ and a PPT adversary $\mathcal{A}$:*

1. **Setup**: *$\mathcal{C}$ generates a master key MK, a revocation list RL, a state ST, and public parameters PP by running Setup$(1^\lambda)$. It keeps MK, RL to itself and gives PP to $\mathcal{A}$.*

2. **Phase 1**: *$\mathcal{A}$ adaptively request a polynomial number of queries. These queries are processed as follows:*

   - *If this is a private key query for an identity ID, then it gives the corresponding private key $SK_{ID}$ to $\mathcal{A}$ by running GenKey(ID, MK, PP).*

   - *If this is an update key query for time T, then it gives the corresponding update key $UK_{T,R}$ to $\mathcal{A}$ by running UpdateKey(T, RL, MK, PP).*

   - *If this is a decryption key query for an identity ID and time T, then it gives the corresponding decryption key $DK_{ID,T}$ to $\mathcal{A}$ by running DeriveKey($SK_{ID}$, $UK_T$, PP).*

   - *If this is a revocation query for an identity ID and revocation time T, then it updates the revocation list RL by running Revoke(ID, T, RL, ST) with the restriction: The revocation query for time T cannot be queried if the update key query for the time T was already requested. Note that we assume that the update key queries and the revocation queries are requested in non-decreasing order of time.*

3. **Challenge**: *$\mathcal{A}$ submits a challenge identity $ID^*$, challenge time $T^*$, and two challenge messages $M_0^*, M_1^*$ with equal length with the following restrictions:*

   - *If a private key query for an identity ID such that $ID = ID^*$ was requested, then the identity $ID^*$ should be revoked at some time T such that $T \leq T^*$.*

   - *The decryption key query for $ID^*$ and $T^*$ was not requested.*
   *$\mathcal{C}$ flips random $\mu \in \{0, 1\}$ and obtains a ciphertext $CT^*$ by running Encrypt $(ID^*, T^*, M_\mu^*, PP)$. It gives $CT^*$ to $\mathcal{A}$.*

4. **Phase 2**: *$\mathcal{A}$ may continue to request a polynomial number of additional queries subject to the same restrictions as before.*

5. **Guess**: *Finally, $\mathcal{A}$ outputs a guess $\mu' \in \{0, 1\}$, and wins the game if $\mu = \mu'$.*

   *The advantage of $\mathcal{A}$ is defined as $Adv_{\mathcal{A}}^{RIBE}(\lambda) = |\Pr[\mu = \mu'] - \frac{1}{2}|$ where the probability is taken over all the randomness of the experiment. An RIBE scheme is IND-CPA secure if for all PPT adversary $\mathcal{A}$, the advantage of $\mathcal{A}$ is negligible in the security parameter $\lambda$.*

## 3 Revocable IBE with SD

In this section, we first review the perfect binary tree and the subset difference method, and then we propose a generic construction for RIBE by combining subset difference, IBE, IBR, and HIBE schemes.

## 3.1 Binary tree

A perfect binary tree $\mathcal{BT}$ is a tree data structure in which all internal nodes have two child nodes and all leaf nodes have the same depth. Let $N = 2^n$ be the number of leaf nodes in $\mathcal{BT}$. The number of all nodes in $\mathcal{BT}$ is $2N - 1$ and we denote $v_i$ as a node in $\mathcal{BT}$ for any $1 \leq i \leq 2N - 1$. The depth $d_i$ of a node $v_i$ is the length of the path from a root node to the node. The root node of a tree has depth zero. The depth of $\mathcal{BT}$ is the length of the path from the root node to a leaf node. A level of $\mathcal{BT}$ is a set of all nodes at given depth.

Each node $v_i \in \mathcal{BT}$ has an identifier $L_i \in \{0, 1\}^*$ which is a fixed and unique string. An identifier of each node is assigned as follows: Each edge in the tree is assigned with 0 or 1 depending on whether it is connected to the left or right child node. The identifier $L_i$ of a node $v_i$ is obtained by reading all labels of edges in a path from the root node to the node $v_i$. The root node has an empty identifier $\epsilon$. For a node $v_i$, we define $Label(v_i)$ be the identifier of $v_i$ and $Depth(v_i)$ be the depth $d_i$ of $v_i$.

A subtree $\mathcal{T}_i$ in $\mathcal{BT}$ is defined as a tree that is rooted at a node $v_i \in \mathcal{BT}$. A subset $S_i$ is defined as a set of all leaf nodes in $\mathcal{T}_i$. For any two nodes $v_i, v_j \in \mathcal{BT}$ where $v_j$ is a descendant of $v_i$, $\mathcal{T}_{i,j}$ is defined as a subtree $\mathcal{T}_i - \mathcal{T}_j$, that is, all nodes that are descendants of $v_i$ but not $v_j$. A subset $S_{i,j}$ is defined as the set of leaf nodes in $\mathcal{T}_{i,j}$, that is, $S_{i,j} = S_i \backslash S_j$.

For a perfect binary tree $\mathcal{BT}$ and a subset $R$ of leaf nodes, $ST(\mathcal{BT}, R)$ is defined as the Steiner Tree induced by the set $R$ and the root node, that is, the minimal subtree of $\mathcal{BT}$ that connects all the leaf nodes in $R$ and the root node.

## 3.2 Subset difference method

The subset difference (SD) method is one instance of the subset cover (SC) framework proposed by Naor et al. [19] which was used for efficient symmetric key broadcast encryption. The SD method is more efficient than the complete subtree (CS) method because the size of the cover set representing the non-revoked users is smaller than that of the CS method. We follow the SD definition of Lee et al. [36]. The SD method uses a perfect binary tree and each user is located at a leaf node in the binary tree. The **Assign** algorithm computes a path set $PV$, which is consists of subsets associated with the path from the root node to a user's leaf node. The **Cover** algorithm derives a cover set $CV$ that can effectively cover non-revoked leaf nodes. The **Match** algorithm can derive two related subsets if a user's leaf node is not revoked in the cover set. A simple example of the SD method is given in Figs 1 and 2. A detailed description of the SD method is given as follows.

**SD.Setup**($N$): Let $N = 2^n$ be the number of leaf nodes. It sets a perfect binary tree $\mathcal{BT}$ of depth $n$ and outputs $\mathcal{BT}$. Note that a user is assigned to a leaf node in $\mathcal{BT}$ and the collection $\mathcal{S}$ of SD is the set of all subsets $\{S_{i,j}\}$ where $v_i, v_j \in \mathcal{BT}$ and $v_j$ is a descendant of $v_i$.

**SD.Assign**($\mathcal{BT}, v$): Let $v$ be the leaf node of $\mathcal{BT}$ that is assigned to a user $ID$. Let $(v_{k_0}, v_{k_1}, \ldots, v_{k_n})$ be a path from the root node $v_{k_0}$ to the leaf node $v_{k_n} = v$. It initializes a path set $PV$ as an empty one. For all $i, j \in \{k_0, \ldots, k_n\}$ such that $v_j$ is a descendant of $v_i$, it adds a subset $S_{i,j}$ defined by two nodes $v_i$ and $v_j$ in the path into $PV$. It outputs the path set $PV = \{S_{i,j}\}$.

**SD.Cover**($\mathcal{BT}, R$): Let $R$ be a revoked set of leaf nodes (or users). It first sets a subtree $\mathcal{T}$ as $ST(\mathcal{BT}, R)$, and then it builds a cover set $CV$ iteratively by removing nodes from $\mathcal{T}$ until $\mathcal{T}$ consists of just a single node as follows:

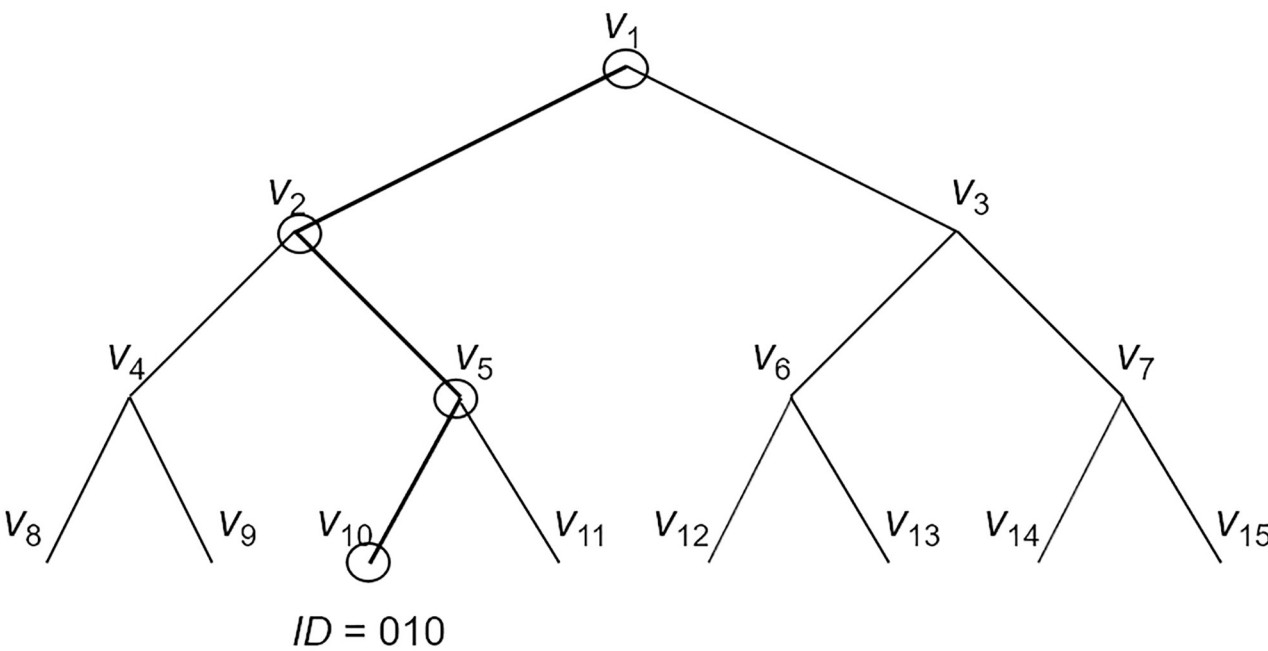

$$ID = 010$$

$$PV_{ID} = \{(v_1, v_2),\ (v_1, v_5),\ (v_1, v_{10}),\ (v_2, v_5),\ (v_2, v_{10}),\ (v_5, v_{10})\}$$

**Fig 1. A path set for *ID* = 010 in the SD method.**

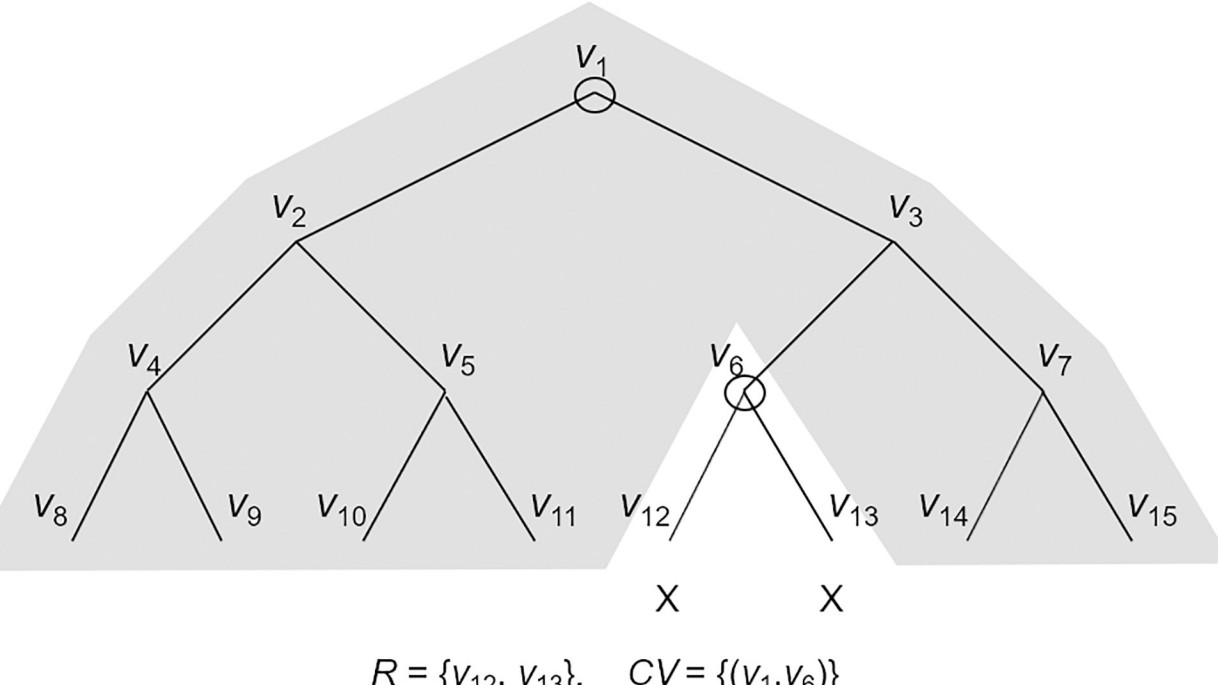

$$R = \{v_{12}, v_{13}\}, \quad CV = \{(v_1, v_6)\}$$

**Fig 2. A cover set for *R* = {*v*₁₂, *v*₁₃} in the SD method.**

1. It finds two leaf nodes $v_i$ and $v_j$ in $\mathcal{T}$ such that the least-common-ancestor $v$ of $v_i$ and $v_j$ does not contain any other leaf nodes of $\mathcal{T}$ in its subtree. Let $v_l$ and $v_k$ be the two child nodes of $v$ such that $v_i$ is a descendant of $v_l$ and $v_j$ is a descendant of $v_k$. If there is only one leaf node left, it makes $v_i = v_j$ to the leaf node, $v$ to be the root of $\mathcal{T}$ and $v_l = v_k = v$.

2. If $v_l \neq v_i$, then it adds the subset $S_{l,i}$ to $CV$; likewise, if $v_k \neq v_j$, it adds the subset $S_{k,j}$ to $CV_R$.

3. It removes from $\mathcal{T}$ all the descendants of $v$ and makes $v$ a leaf node.
   It outputs the cover set $CV = \{S_{i,j}\}$.

**SD.Match**($CV$, $PV$): Let $CV = \{S_{i,j}\}$ and $PV = \{S_{i,j}\}$. It finds two subsets $S_{i,j} \in CV$ and $S_{i', j'} \in PV$ such that $(v_i = v_{i'}) \wedge (d_j = d_{j'}) \wedge (v_j \neq v_{j'})$ where $d_j$ is the depth of $v_j$. If two subsets exist, then it outputs $(S_{i,j}, S_{i', j'})$. Otherwise, it outputs $\perp$.

The correctness of the SD scheme requires that if $v \notin R$, then **SD.Match**($CV$, $PV$) = $(S_{i,j}, S_{i', j'})$ such that $(v_i = v_{i'}) \wedge (d_j = d_{j'}) \wedge (v_j \neq v_{j'})$ where $S_{i,j}$ is defined by two nodes $v_i$ and $v_j$.

**Lemma 3.1** ([19]). *Let $N = 2^n$ be the number of leaf nodes in a perfect binary tree and $r$ be the size of a revoked set. In the SD method, the size of a path set is $O(\log^2 N)$ where the hidden constant is $1/2$ and the size of a cover set is at most $2r - 1$.*

## 3.3 Design principle

In order to design a generic RIBE scheme with the SD method, we first analyze the generic RIBE scheme with the CS method proposed by Ma and Lin [18]. The key design principle of their RIBE scheme with the CS method is that the identity $ID$ of a receiver can be fixed to the path of a binary tree and a ciphertext is associated with the path set of the receiver's identity $ID$ where as the private key of a user is associated with the path set of a binary tree in directly constructed many RIBE schemes. Therefore, if the receiver's identity $ID$ is not revoked in the CS method, there is a common node in the path set of the binary tree and a node in the cover set of an update key. Thus, the equality function of IBE can be used to handle this common node since the path can be related to IBE ciphertexts and the cover set can be related to IBE private keys.

However, this design method is difficult to apply to the SD method. The reason is that in the SD method, unlike the CS method, there are no common nodes in the path set and the cover set. To solve this problem, we use the new interpretation of the SD method which was used for an efficient public-key revocation (PKR) scheme and RIBE scheme by using the SD method [15, 36]. To design an efficient PKR scheme, Lee et al. [36] observed that the subset $S_{i,j}$ of the SD method can be interpreted as a set of single member revocation instead of the existing interpretation that the subset $S_{i,j}$ is a set of leaf nodes where each leaf node belongs to the subtree $\mathcal{T}_i$ but does not belong to the subtree $\mathcal{T}_j$. That is, if we consider a group set $GL$ which consists of all nodes of the subtree $\mathcal{T}_i$ that has the same depth as the node $v_j$, the subset $S_{i,j}$ can be interpreted as the same as $GL$ except that the node $v_j$ is excluded from $GL$. Thus, $S_{i,j}$ can be interpreted as single member revocation because it revokes one node $v_j$ in $GL$.

This interesting observation was also used to directly construct an RIBE scheme with the SD method by Lee et al. [15]. They used a degree-one polynomial in the exponent to implement single member revocation, but they only achieved an RIBE scheme in a non-generic way. In this work, we found that IBE and IBR schemes can be combined in a generic way to achieve single member revocation if an RIBE ciphertext is associated with a path set $PV$ for a receiver's identity $ID$ and an RIBE update key is associated with a cover set $CV$ for a revoked set $R$. That is, given the subset $S_{i,j}$, if we set a group label $GL = L_i \| d_j$ and a member label $ML = L_j$ where $L_i$,

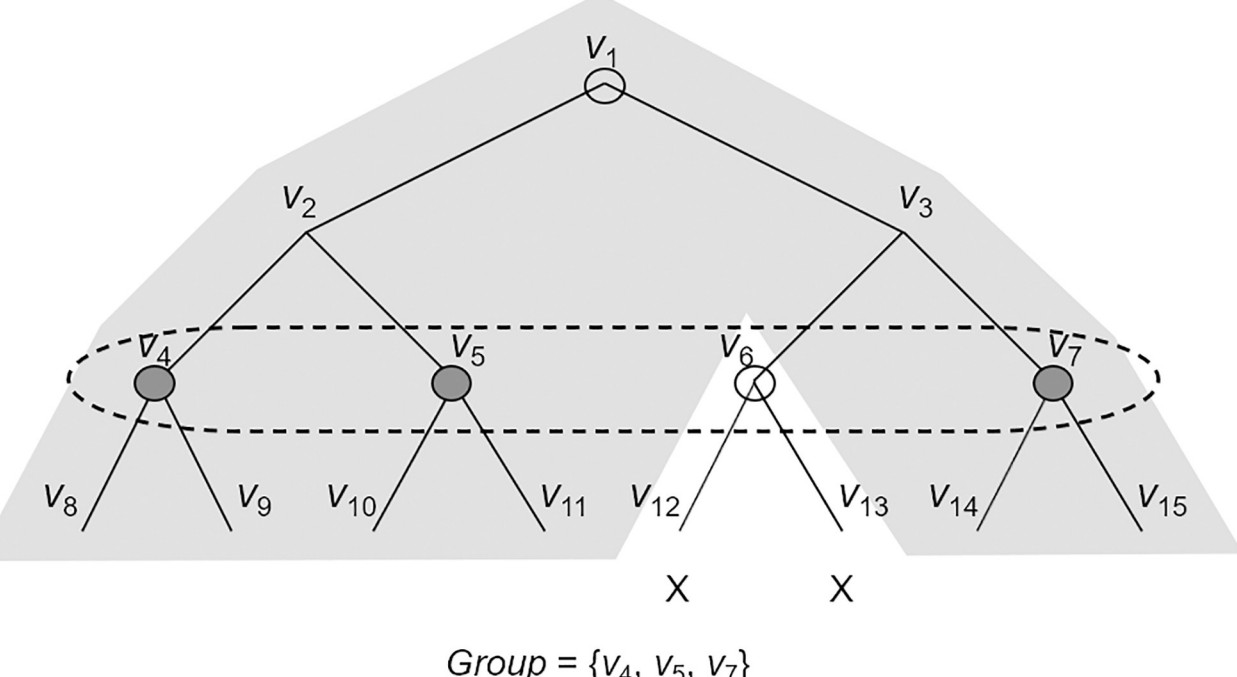

**Fig 3. A single member revoked group from the subset $S_{1,6} = (v_1, v_6)$.**

$L_j$ are identifiers of nodes $v_i$, $v_j$ and $d_j$ is the depth of $v_j$, then all members of the group $GL$ can be represented by a label pair $(GL, ML)$. In this case, a label pair $(GL, ML)$ in a ciphertext and another label pair $(GL', ML')$ in an update key can be matching pairs if the group labels are equal but the member labels are different such that $GL = GL' \wedge ML \neq ML'$. Thus, we can support the equality $GL = GL'$ by using an IBE scheme, and we can support the inequality $ML \neq ML'$ by using an IBR scheme. In addition, to provide security against collusion attacks in the black-box construction, we divided the message $M$ of a ciphertext into several secret shares by using a simple secret sharing scheme, and then encrypt these shares by using IBE and IBR schemes. Additionally, we use an HIBE scheme to provide the decryption key exposure resistance.

### 3.4 Generic construction

Let **IBE** = (**Setup**, **GenKey**, **Encrypt**, **Decrypt**) be an IBE scheme, **IBR** = (**Setup**, **GenKey**, **Encrypt**, **Decrypt**) be an IBR scheme that supports a single revoked identity, and **HIBE** = (**Setup**, **GenKey**, **Delegate**, **Encrypt**, **Decrypt**) be a two-level HIBE scheme. We define $GMLabels(S_{i,j}) = (GL = Label(v_i) \| Depth(v_j), ML = Label(v_j))$ where $GL$ is a group label and $ML$ is a member label. A simple example of a group of nodes derived from a subset $S_{i,j}$ is given in Fig 3. A generic RIBE scheme using the SD method is described as follows.

**RIBE.Setup**($1^\lambda$): Let $\mathcal{I} = \{0, 1\}^n$ be the identity space.

1. It first obtains $MK_{IBE}$, $PP_{IBE}$ by running **IBE.Setup**($1^\lambda$). It obtains $MK_{IBR}$, $PP_{IBR}$ by running **IBR.Setup**($1^\lambda$). It also obtains $MK_{HIBE}$, $PP_{HIBE}$ by running **HIBE.Setup**($1^\lambda$, 2).

2. It defines a binary tree $\mathcal{BT}$ by running **SD.Setup**($2^n$) where $\mathcal{I} \in \{0, 1\}^n$. Note that it will deterministically assign an identity $ID$ to a leaf node $v \in \mathcal{BT}$ such that $Label(v) = ID$.

3. It outputs a master key $MK = (MK_{IBE}, MK_{IBR}, MK_{HIBE})$, a revocation list $RL = \emptyset$, and public parameters $PP = (PP_{IBE}, PP_{IBR}, PP_{HIBE}, \mathcal{BT})$.

**RIBE.GenKey**($ID$, $MK$, $PP$): To generate a private key for $ID$, it proceeds as follows:

1. It obtains $SK_{HIBE}$ by running **HIBE.GenKey**(($ID$), $MK_{HIBE}$, $PP_{HIBE}$).

2. Finally, it outputs a private key $SK_{ID} = SK_{HIBE}$.

**RIBE.UpdateKey**($T$, $RL$, $MK$, $PP$): To generate an update key for $T$, it proceeds as follows:

1. It initializes $RV = \emptyset$. For each $(ID_j, T_j) \in RL$, it adds a leaf node $v_j \in \mathcal{BT}$ which is associated with $ID_j$ into $RV$ if $T_j \leq T$. It obtains $CV_T$ by running **SD.Cover** $(\mathcal{BT}, RV)$.

2. For each $S_{i,j} \in CV_T$, it sets labels $(GL, ML) = GMLabels(S_{i,j})$ and performs: It obtains $SK_{IBE,S_{i,j}}$ by running **IBE.GenKey**($GL \| T$, $MK_{IBE}$, $PP_{IBE}$) and $SK_{IBR,S_{i,j}}$ by running **IBR. GenKey** ($ML \| T$, $MK_{IBR}$, $PP_{IBR}$).

3. Finally, it outputs an update key $UK_T = (CV_T, \{(SK_{IBE,S_{i,j}}, SK_{IBR,S_{i,j}})\}_{S_{i,j} \in CV_T})$.

**RIBE.DeriveKey**($SK_{ID}$, $UK_T$, $PP$): Let $SK_{ID} = SK_{HIBE}$. To derive a decryption key for $ID$ and $T$, it proceeds as follows:

1. It obtains $DK_{HIBE}$ by running **HIBE.Delegate**(($ID$, $T$), $SK_{HIBE}$, $PP_{HIBE}$).

2. Finally, it outputs a decryption key $DK_{ID,T} = (UK_T, DK_{HIBE})$.

**RIBE.Encrypt**($ID$, $T$, $M$, $PP$): To generate a ciphertext for $ID$ and $T$, it proceeds as follows:

1. It selects random $R_1$ and sets $R_2 = M \oplus R_1$. It obtains $CT_{HIBE}$ by running **HIBE.Encrypt** (($ID$, $T$), $R_2$, $PP_{HIBE}$).

2. Let $v_{ID}$ be a leaf node associated with $ID$ such that $ID = Label(v_{ID})$. Recall that the leaf node $v_{ID}$ is fixed by the $Label$ function. It obtains $PV_{ID}$ by running **SD.Assign** $(\mathcal{BT}, v_{ID})$.

3. For each $S_{i,j} \in PV_{ID}$, it sets labels $(GL, ML) = GMLabels(S_{i,j})$ and performs:

   1. It selects random $R_{3,S_{i,j}}$ and sets $R_{4,S_{i,j}} = R_1 \oplus R_{3,S_{i,j}}$.

   2. It obtains $CT_{IBE,S_{i,j}}$ by running **IBE.Encrypt** ($GL \| T$, $R_{3,S_{i,j}}$, $PP_{IBE}$) and obtains $CT_{IBR,S_{i,j}}$ by running **IBR.Encrypt** ($ML \| T$, $R_{4,S_{i,j}}$, $PP_{IBR}$).
   It creates $CT_{PV} = (PV_{ID}, \{(CT_{IBE,S_{i,j}}, CT_{IBR,S_{i,j}})\}_{S_{i,j} \in PV_{ID}})$.

4. Finally, it outputs a ciphertext $CT_{ID,T} = (CT_{PV}, CT_{HIBE})$.

**RIBE.Decrypt**($CT_{ID,T}$, $DK_{ID', T'}$, $PP$): Let $CT_{ID,T} = (CT_{PV}, CT_{HIBE})$ and $DK_{ID', T'} = (UK_T, DK_{HIBE})$. It proceeds as follows:

1. It first obtains $R_2$ by running **HIBE.Decrypt**($CT_{HIBE}$, $DK_{HIBE}$, $PP_{HIBE}$).

2. It finds $(S_{i,j}, S_{i', j'}) = $ **SD.Match**($PV_{ID}$, $CV_T$). It retrieves $CT_{IBE,S_{i,j}}$, $CT_{IBR,S_{i,j}}$ from $CT_{PV}$ and $SK_{IBE,S_{i',j'}}$, $[0]SK_{IBR,S_{i',j'}}$ from $UK_T$. Next, it obtains $R_3$ by running **IBE.Decrypt** ($CT_{IBE,S_{i,j}}$, $SK_{IBE,S_{i',j'}}$, $PP_{IBE}$) and obtains $R_4$ by running **IBR.Decrypt** ($CT_{IBR,S_{i,j}}$, $SK_{IBR,S_{i',j'}}$, $PP_{IBR}$).

3. Finally, it outputs a message $M = R_3 \oplus R_4 \oplus R_2$.

**RIBE.Revoke**(*ID*, *T*, *RL*): If (*ID*, *) already exists in *RL*, it outputs *RL*. Otherwise, it adds (*ID*, *T*) to *RL* and outputs the updated *RL*.

## 3.5 Correctness

The correctness of the above RIBE scheme can be easily seen by using the correctness of the underlying IBE, IBR, HIBE and SD schemes. Let $CT_{ID,T} = (CT_{PV}, CT_{HIBE})$ be a ciphertext associated with *ID* and *T*, and $DK_{ID', T'} = (UK_T, DK_{HIBE})$ be a decryption key is associated with *ID'* and *T'*. In this case, if the condition $ID = ID' \wedge T = T'$ is satisfied, then the random $R_2$ is correctly decrypted by running **HIBE.Decrypt**($CT_{HIBE}$, $SK_{HIBE}$, $PP_{HIBE}$) because of the correctness of HIBE.

Now we show that random $R_1$ can be correctly decrypted from $CT_{PV}$ and $UK_T$ if the identity *ID* of the ciphertext is not revoked in the update key $UK_T$. Recall that the ciphertext $CT_{PV}$ is associated with $PV_{ID}$ and the update key $UK_T$ is associated with $CV_T$. By the correctness of the SD scheme, the **SD.Match** algorithm outputs two subsets of $S_{i,j}$, $S_{i', j'}$ such that $(v_i = v_{i'}) \wedge (d_j = d_{j'}) \wedge (v_j \neq v_{j'})$ if the leaf node $v_{ID}$ is not included in the revoked set *RV*. Let $(CT_{IBE,S_{i,j}}, CT_{IBR,S_{i,j}}) \in CT_{PV}$ and $(SK_{IBE,S_{i',j'}}, SK_{IBR,S_{i',j'}}) \in UK_T$ be corresponding tuples of $S_{i,j}$ and $S_{i', j'}$ respectively. From the definition of *GMLabels*, labels $(GL, ML) = GMLabels(S_{i,j})$ and $(GL', ML') = GMLabels(S_{i', j})$ are obtained and they satisfy $GL = GL' \wedge ML \neq ML'$. Therefore, if the time *T* of the ciphertext is the same as the time *T'* of the update key, then random $R_3$ can be decrypted by running **IBE.Decrypt** $(CT_{IBE,S_{i,j}}, SK_{IBE,S_{i',j'}}, PP_{IBE})$ because of $GL \| T = GL' \| T'$ and random $R_4$ can be decrypted by running **IBR.Decrypt** $(CT_{IBR,S_{i,j}}, SK_{IBR,S_{i',j'}}, PP_{IBR})$ because of $ML \| T \neq ML' \| T'$. Therefore, random $R_1$ is obtained from $R_3 \oplus R_4$.

## 3.6 Discussions

**Layered subset difference.**   Since our generic RIBE scheme uses the SD method, the size of a ciphertext depends on the size of the *PV* set and the size of an update key depends on the size of the *CV* set in the SD method. Thus, the ciphertext and update key of generic RIBE consists of approximately $O(\log^2 N)$ IBE ciphertexts and $2r$ IBE private keys respectively where $N = 2^n$ is the number of users and *r* is the number of revoked users. In order to reduce the size of ciphertexts in this generic RIBE scheme, we can apply the layered subset difference (LSD) method of Halevy and Shamir [20]. If the LSD method is used instead of the SD method, the ciphertext and the update key of this general RIBE scheme consists of $O(\log^{1.5} N)$ IBE ciphertexts and $4r$ IBE private keys, respectively.

**Chosen-ciphertext security.**   The CCA security model, which is stronger than the CPA security model, allows an adversary to request decryption queries on ciphertexts. The above generic RIBE construction only can derive a CPA secure RIBE scheme by using CPA secure IBE, IBR, and HIBE schemes as building blocks. To derive a CCA secure RIBE scheme, we may try to use CCA secure encryption primitives as building blocks. However, this simple construction can not be CCA secure because it allows ciphertext elements reordering attacks. To solve this problem, we apply the CCA methodology for multiple encryption proposed by Dodis and Katz [37]. That is, a CCA secure RIBE scheme can be constructed by combining CCA secure IBE, IBR, HIBE schemes with a one-time signature (OTS) scheme with strong unforgeability. At this time, the underlying IBE, IBR, and HIBE schemes should be modified to receive additional labels as inputs since the public key of OTS should be tied with ciphertexts. This approach also provides the decryption key exposure resistance (DKER) property since a decryption key is generated by using the delegation property of HIBE.

# 4 Security analysis

In this section, we prove the IND-CPA security of the generic RIBE construction proposed in the previous section. The basic idea of this proof is to show that if there is an attacker that breaks the IND-CPA security of the RIBE scheme, then we can construct an algorithm that breaks the IND-CPA security of underlying IBE, IBR, or HIBE schemes. In order to simplify the security proof, we try to prove the security by separating the attacker into two types. That is, the Type-I attacker does not request a private key query on the challenge identity $ID^*$, and the Type-II attacker requests a private key query on the identity $ID^*$.

First, since the Type-I attacker does not query the private key for the identity $ID^*$, we perform the proof that relates the security of the underlying HIBE scheme with the security of the RIBE scheme. Next, since the Type-II attacker queries the private key for $ID^*$, we perform the proof that relates the security of the underlying IBE or IBR scheme and the security of the RIBE scheme.

**Theorem 4.1**. *The generic RIBE scheme is IND-CPA secure if the underlying IBE, IBR, and HIBE schemes are IND-CPA secure.*

*Proof.* Let $ID^*$ be the challenge identity and $T^*$ be the challenge time. We divide the behavior of an adversary as two types: Type-I and Type-II, which are defined as follows:

**Type-I**. An adversary is Type-I if it requests a private key for $ID \neq ID^*$ for all private key queries. In this case, the adversary can request a decryption key for $ID$ and $T$ such that $ID \neq ID^*$ or $ID = ID^* \wedge T \neq T^*$.

**Type-II**. An adversary is Type-II if it requests a private key for $ID = ID^*$ for some private key query. In this case, the private key for $ID^*$ should be revoked at some time $T$ such that $T \leq T^*$ by the restriction of the security model.

Let $E_i$ be the event that $\mathcal{A}$ behaves like Type-i adversary. From Lemmas 4.2 and 4.3, we obtain the following result

$$\begin{aligned}
\mathbf{Adv}_{\mathcal{A}}^{RIBE}(\lambda) &\leq \Pr[E_I] \cdot \mathbf{Adv}_{\mathcal{A}}^{RIBE}(\lambda) + \Pr[E_{II}] \cdot \mathbf{Adv}_{\mathcal{A}}^{RIBE}(\lambda) \\
&\leq \mathbf{Adv}_{\mathcal{B}}^{HIBE}(\lambda) + O(n^2)\big(\mathbf{Adv}_{\mathcal{B}}^{IBE}(\lambda) + \mathbf{Adv}_{\mathcal{B}}^{IBR}(\lambda)\big).
\end{aligned}$$

This completes our proof.

## 4.1 Type-I adversary

The Type-I attacker does not request a private key query on the challenge $ID^*$, but can request decryption key queries such that $ID = ID^*$ and $T \neq T^*$. To deal with this attacker, we build a reduction algorithm that attacks the HIBE scheme and selects the other IBE and IBR schemes by itself. In this case, this algorithm will be able to handle all queries of the Type-I attacker by using the queries for the HIBE scheme. The detailed proof is as follows.

**Lemma 4.2**. *For the Type-I adversary, the generic RIBE scheme is IND-CPA secure if the HIBE scheme is IND-CPA secure.*

*Proof.* Suppose there exists an adversary $\mathcal{A}$ that attacks the RIBE scheme with a non-negligible advantage. An algorithm $\mathcal{B}$ that attacks the HIBE scheme is initially given public parameters $PP_{HIBE}$ by a challenger $\mathcal{C}$. Then $\mathcal{B}$ that interacts with $\mathcal{A}$ is described as follows:

**Setup**: $\mathcal{B}$ generates $MK_{IBE}, PP_{IBE}$ by running the **IBE.Setup** algorithm, generates $MK_{IBR}$, $PP_{IBR}$ by running the **IBR.Setup** algorithm. It initializes $RL = \emptyset$ and gives $PP = (PP_{IBE}, PP_{IBR}, PP_{HIBE})$ to $\mathcal{A}$.

**Phase 1**: $\mathcal{A}$ adaptively requests a polynomial number of private key, update key, decryption key, and revocation queries.

- For a private key query with an identity $ID$, $\mathcal{B}$ proceeds as follows: It receives $SK_{HIBE}$ from $\mathcal{C}$ by querying a private key for $ID$ since $ID \neq ID^*$ by the restriction of the Type-I adversary. It gives $SK_{ID} = SK_{HIBE}$ to $\mathcal{A}$.

- For an update key query with time $T$, $\mathcal{B}$ proceeds as follows: It simply generates $UK_T$ by running the **RIBE.UpdateKey** algorithm since it knows $MK_{IBE}$ and $MK_{IBR}$. It gives $UK_T$ to $\mathcal{A}$.

- For a decryption key query with an identity $ID$ and time $T$, $\mathcal{B}$ proceeds as follows:

  1. It generates $UK_T$ by running the **RIBE.UpdateKey** algorithm since it knows $MK_{IBE}$ and $MK_{IBR}$.

  2. It receives $DK_{HIBE}$ from $\mathcal{C}$ by querying a private key for $ID$ and $T$ since $ID \neq ID^*$ or $ID = ID^* \wedge T \neq T^*$ by the restriction of the Type-I adversary.

  3. It gives $DK_{ID,T} = (UK_T, DK_{HIBE})$ to $\mathcal{A}$.

- For a revocation query with an identity $ID$ and time $T$, $\mathcal{B}$ proceeds as follows: It adds $(ID, T)$ to $RL$ if $ID$ was not revoked before.

  **Challenge**: $\mathcal{A}$ submits a challenge identity $ID^*$, challenge time $T^*$, and two challenge messages $M_0^*, M_1^*$. $\mathcal{B}$ proceeds as follows:

1. It first select random $R_1$ and sets $R_{2,0} = M_0^* \oplus R_1, R_{2,1} = M_1^* \oplus R_1$.

2. Next, it receives $CT_{HIBE}^*$ from $\mathcal{C}$ by submitting $ID^*$, $T^*$, and two challenge messages $R_{2,0}, R_{2,1}$.

3. To creates $CT_{PV}^*$ for $ID^*$ and $T^*$, it simply follows the procedures in the **RIBE.Encrypt** algorithm with the random $R_1$ input.

4. It gives a challenge ciphertext $CT^* = (CT_{PV}^*, CT_{HIBE}^*)$ to $\mathcal{A}$.

   **Phase 2**: Same as Phase 1.
   **Guess**: Finally, $\mathcal{A}$ outputs a guess $\mu' \in \{0, 1\}$. $\mathcal{B}$ also outputs $\mu'$.

## 4.2 Type-II adversary

Since the Type-II attacker requests a private key query on the challenge $ID^*$, we can not handle the private key queries of the RIBE scheme by using the private key queries of the HIBE scheme in the proof. Therefore, we prove the security by relating the security of the IBE and IBR schemes with the security of the RIBE scheme against the Type-II attacker.

The main idea of the proof is to take advantage of the restriction of the RIBE security model such that if the attacker queries the private key for the challenge identity $ID^*$, then the corresponding private key for $ID^*$ must be revoked from the update key on the challenge time $T^*$. Thus, the ciphertext $CT_{PV}^*$ in the challenge ciphertext consists of the IBE and IBR ciphertexts associated with the subset $S_{i,j}$ belonging to the path set $PV_{ID^*}$, but the IBE and IBR private keys that can decrypt the corresponding ciphertext elements in $CT_{PV}^*$ are not included in the update key for $T^*$ because of the restriction. Using this fact, we can prove the security of the RIBE scheme against the Type-II attacker by using the security of the IBE or IBR scheme.

We prove the security by using hybrid games consisting of multiple sub-games because the ciphertext $CT_{PV}^*$ is composed of many IBE and IBR ciphertexts. That is, in the hybrid games, a ciphertext which encrypts a random value related to $M_0^*$ is changed to another ciphertext which encrypts a random value related to $M_1^*$. In this hybrid steps, since the number of IBE and IBR ciphertext pairs in $CT_{PV}^*$ is maximum $O(n^2)$, the proof can be completed by performing $O(n^2)$ hybrid games. The detailed proof is described as follows.

**Lemma 4.3**. *For the Type-II adversary, the generic RIBE scheme is IND-CPA secure if the IBE and IBR schemes are IND-CPA secure.*

*Proof.* Let $ID^*$ be the challenge identity and $PV_{ID^*}$ be the path set of $ID^*$ where the number of subsets in $PV_{ID^*}$ is $\ell = n(n-1)/2$. The challenge ciphertext is formed as $CT^* = (CT_{PV}^*, CT_{HIBE}^*)$ where $CT_{PV}^* = (PV_{ID^*}, \{(CT_{IBE,S_{i_k j_k}}^*, CT_{IBR,S_{i_k j_k}}^*)\}_{k=1}^{\ell})$. For the security proof, we define hybrid games $\mathbf{G}_0, \mathbf{G}_1, \mathbf{G}_2, \mathbf{G}_3$ as follows:

**Game $\mathbf{G}_0$**. This game is the original security game defined in the security model except that the challenge bit $\mu$ is fixed to 0.

**Game $\mathbf{G}_1$**. This game is the same as the game $\mathbf{G}_0$ except that the settings of random $R_1$ and $R_2$ in the challenge ciphertext are changed. That is, $R_2$ are randomly chosen and $R_1$ is set as $M_0^* \oplus R_2$.

**Game $\mathbf{G}_2$**. In this game, the generation of $CT_{PV}^*$ in the challenge ciphertext $CT^*$ is changed. That is, a random $R_1' = M_1^* \oplus R_2$ is encrypted instead of $R_1 = M_0^* \oplus R_2$ to generate $CT_{PV}^*$. For the analysis of security, we define additional sub-games $\mathbf{H}_0, \ldots, \mathbf{H}_\rho, \ldots, \mathbf{H}_\ell$ where $\mathbf{H}_0 = \mathbf{G}_1$ and $\mathbf{H}_\ell = \mathbf{G}_2$. The game $\mathbf{H}_\rho$ is similar to the game $\mathbf{H}_{\rho-1}$ except that the tuple $(CT_{IBE,S_{i_\rho j_\rho}}^*, CT_{IBR,S_{i_\rho j_\rho}}^*)$ is an encryption on the random $R_1' = M_1^* \oplus R_2$. Specifically, each tuple $(CT_{IBE,S_{i_k j_k}}^*, CT_{IBR,S_{i_k j_k}}^*)$ for $k \leq \rho$ is an encryption on the random $R_1' = M_1^* \oplus R_2$ and each tuple $(CT_{IBE,S_{i_k j_k}}^*, CT_{IBR,S_{i_k j_k}}^*)$ for $k > \rho$ is an encryption on the random $R_1 = M_0^* \oplus R_2$. In the game $\mathbf{H}_\rho$, we let $(GL^*, ML^*) = GMLabels(S_{i_\rho j_\rho})$ of $S_{i_\rho j_\rho} \in PV_{ID^*}$ where $CT_{PV}^*$ is related with $PV_{ID^*}$. We divide the adversary as the following sub-types:
- An adversary is **Type-II-A** if it requests an update key for time $T^*$ such that $GL \neq GL^*$ for all labels $(GL, ML) = GMLabels(S_{i,j})$ of $S_{i,j} \in CV_R$ where $UK_{T^*}$ is related with $CV_R$.
- An adversary is **Type-II-B** if it requests an update key for time $T^*$ such that $GL = GL^*$ for some labels $(GL, ML) = GMLabels(S_{i,j})$ of $S_{i,j} \in CV_R$ where $UK_{T^*}$ is related with $CV_R$. In this case, we have that $ML \neq ML^*$ since the identity $ID^*$ should be revoked in $UK_{T^*}$ by the restriction of the security model.

**Game $\mathbf{G}_3$**. This game is the same as the game $\mathbf{G}_2$ except that the settings of random $R_1'$ and $R_2$ in the challenge ciphertext are changed. That is, $R_1'$ is randomly chosen and $R_2$ is set as $M_1^* \oplus R_1'$. This game is the original security game in the security model except that the challenge bit $\mu$ is fixed to 1.

Let $S_{\mathcal{A}}^{G_i}$ be the event that $\mathcal{A}$ outputs 0 in a game $\mathbf{G}_i$. From Lemmas 4.4 and 4.5, we obtain the following result

$$
\begin{aligned}
\mathbf{Adv}_{\mathcal{A}}^{RIBE}(\lambda) \quad &\leq \frac{1}{2}|\Pr[S_{\mathcal{A}}^{G_0}] - \Pr[S_{\mathcal{A}}^{G_3}]| \leq \frac{1}{2}|\Pr[S_{\mathcal{A}}^{G_1}] - \Pr[S_{\mathcal{A}}^{G_2}]| \\
&\leq \frac{1}{2}\left(\sum_{\rho=1}^{\ell}|\Pr[S_{\mathcal{A}}^{H_{\rho-1}}] - \Pr[S_{\mathcal{A}}^{H_\rho}]|\right) \leq O(n^2)\left(\mathbf{Adv}_{\mathcal{B}}^{IBE}(\lambda) + \mathbf{Adv}_{\mathcal{B}}^{IBR}(\lambda)\right).
\end{aligned}
$$

This completes our proof.

**Lemma 4.4**. *If the IBE scheme is IND-CPA secure, then no polynomial-time Type-II-A adversary can distinguish between $\mathbf{H}_{\rho-1}$ and $\mathbf{H}_\rho$ with a non-negligible advantage.*

*Proof.* Suppose there exists an adversary $\mathcal{A}$ that attacks the RIBE scheme with a non-negligible advantage. An algorithm $\mathcal{B}$ that attacks the IBE scheme is initially given public parameters $PP_{IBE}$ by a challenger $\mathcal{C}$. Then $\mathcal{B}$ that interacts with $\mathcal{A}$ is described as follows:

**Setup**: $\mathcal{B}$ generates $MK_{IBR}$, $PP_{IBR}$ by running the **IBR.Setup** algorithm and generates $MK_{HIBE}$, $PP_{HIBE}$ by running the **HIBE.Setup** algorithm. It initializes $RL = \emptyset$ and gives $PP = (PP_{IBE}, PP_{IBR}, PP_{HIBE})$ to $\mathcal{A}$.

**Phase 1**: $\mathcal{A}$ adaptively requests a polynomial number of private key, update key, decryption key, and revocation queries.

- For a private key query with an identity $ID$, $\mathcal{B}$ proceeds as follows: It generates $SK_{ID}$ by running the **RIBE.GenKey** algorithm since it knows $MK_{HIBE}$. It gives $SK_{ID}$ to $\mathcal{A}$.

- For an update key query with time $T$, $\mathcal{B}$ proceeds as follows:

  1. It initializes $RV = \emptyset$. For each $(ID_j, T_j) \in RL$, it adds a leaf node $\nu_j \in \mathcal{BT}$ into $RV$ if $T_j \leq T$. It obtains $CV_T$ by running **SD.Cover** $(\mathcal{BT}, RV)$.

  2. For each $S_{i,j} \in CV_T$, it sets $(GL_k, ML_k) = GMLabels(S_{i,j})$ and performs: It receives $SK_{IBE,S_{i,j}}$ from $\mathcal{C}$ by submitting an identity $GL_k\|T$. It generates $SK_{IBR,S_{i,j}}$ by running **IBR.GenKey** $(ML_k\|T, MK_{IBR}, PP_{IBR})$.

  3. It creates $UK_T = (CV_T, \left\{ (SK_{IBE,S_{i,j}}, SK_{IBR,S_{i,j}}) \right\}_{S_{i,j} \in CV_T})$ and gives $UK_T$ to $\mathcal{A}$.

- For a decryption key query with an identity $ID$ and time $T$, $\mathcal{B}$ proceeds as follows:

  1. It retrieves $SK_{ID} = SK_{HIBE}$ by querying a private key to its own oracle. It also retrieves $UK_T$ by querying an update key to its own oracle.

  2. Next, it generates a delegated key $DK_{HIBE}$ of $SK_{HIBE}$ by running **HIBE.DelegateKey** for $ID$ and $T$.

  3. It gives $DK_{ID,T} = (UK_T, DK_{HIBE})$ to $\mathcal{A}$.

- For a revocation query with an identity $ID$ and time $T$, $\mathcal{B}$ proceeds as follows: It adds $(ID, T)$ to $RL$ if $ID$ was not revoked before.

**Challenge**: $\mathcal{A}$ submits a challenge identity $ID^*$, challenge time $T^*$, and two challenge messages $M_0^*, M_1^*$. $\mathcal{B}$ proceeds as follows:

1. It first selects random $R_2$ and sets $R_{1,M_0} = M_0^* \oplus R_2$, $R_{1,M_1} = M_1^* \oplus R_2$.

2. Next, it generates $CT_{HIBE}^*$ by running **HIBE.Encrypt**$((ID^*, T^*), R_2, PP_{HIBE})$.

3. It obtains $PV_{ID^*}$ by running **SD.Assign** $(\mathcal{BT}, \nu_{ID^*})$ where a leaf node $\nu_{ID^*}$ is associated with $ID^*$. For each $S_{i,j} \in PV_{ID^*}$, it obtains $(GL_k, ML_k) = GMLabels(S_{i,j})$ and proceeds as follows:

   - If $k < \rho$, then it selects random $R_{3,k}$ and sets $R_{4,k} = R_{1,M_1} \oplus R_{3,k}$, and then generates $CT_{IBE,S_{i,j}}^*$ by running **IBE.Encrypt**$(GL_k\|T^*, R_{3,k}, PP_{IBE})$ and $CT_{IBR,S_{i,j}}^*$ by running **IBR.Encrypt** $(ML_k\|T^*, R_{4,k}, PP_{IBR})$.

   - If $k = \rho$, then it performs the follows:

     (a) It selects random $R_{4,k}$ and sets $R_{3,k,M_0} = R_{1,M_0} \oplus R_{4,k}$, $R_{3,k,M_1} = R_{1,M_1} \oplus R_{4,k}$.

     (b) It receives $CT_{IBE,S_{i,j}}^*$ from $\mathcal{C}$ by submitting a challenge identity $GL_k\|T^*$ and challenge messages $R_{3,k,M_0}, R_{3,k,M_1}$.

     (c) It generates $CT_{IBR,S_{i,j}}^*$ by running **IBR.Encrypt**$(ML_k\|T^*, R_{4,k}, PP_{IBR})$.

- If $k > \rho$, then it selects random $R_{3,k}$ and sets $R_{4,k} = R_{1,M_0} \oplus R_{3,k}$, and then generates $CT^*_{IBE,S_{i,j}}$ by running **IBE.Encrypt**$(GL_k \| T^*, R_{3,k}, PP_{IBE})$ and $CT^*_{IBR,S_{i,j}}$ by running **IBR.Encrypt** $(ML_k \| T^*, R_{4,k}, PP_{IBR})$.

  It creates $CT^*_{PV} = (PV_{ID^*}, \{(CT^*_{IBE,S_{i,j}}, CT^*_{IBR,S_{i,j}})\}_{S_{i,j} \in PV_{ID^*}})$.

4. It gives a challenge ciphertext $CT^* = (CT^*_{PV}, CT^*_{HIBE})$ to $\mathcal{A}$.

> **Phase 2**: Same as Phase 1.
>
> **Guess**: Finally, $\mathcal{A}$ outputs a guess $\mu' \in \{0, 1\}$. $\mathcal{B}$ also outputs $\mu'$.
>
> **Lemma 4.5**. *If the IBR scheme is IND-CPA secure, then no polynomial-time Type-II-B adversary can distinguish between* $H_{\rho-1}$ *and* $H_\rho$ *with a non-negligible advantage.*
>
> *Proof.* Suppose there exists an adversary $\mathcal{A}$ that attacks the RIBE scheme with a non-negligible advantage. An algorithm $\mathcal{B}$ that attacks the IBR scheme is initially given public parameters $PP_{IBR}$ by a challenger $\mathcal{C}$. Then $\mathcal{B}$ that interacts with $\mathcal{A}$ is described as follows:
>
> **Setup**: $\mathcal{B}$ generates $MK_{IBE}, PP_{IBE}$ by running the **IBE.Setup** algorithm and generates $MK_{HIBE}, PP_{HIBE}$ by running the **HIBE.Setup** algorithm. It initializes $RL = \emptyset$ and gives $PP = (PP_{IBE}, PP_{IBR}, PP_{HIBE})$ to $\mathcal{A}$.
>
> **Phase 1**: $\mathcal{A}$ adaptively requests a polynomial number of private key, update key, decryption key, and revocation queries.

- For a private key query with an identity $ID$, $\mathcal{B}$ proceeds as follows: It generates $SK_{ID}$ by running **RIBE.GenKey** algorithm since it knows $MK_{HIBE}$. It gives $SK_{ID}$ to $\mathcal{A}$.

- For an update key query with time $T$, $\mathcal{B}$ proceeds as follows:

  1. It initializes $RV = \emptyset$. For each $(ID_j, T_j) \in RL$, it adds a leaf node $v_j \in \mathcal{BT}$ into $RV$ if $T_j \leq T$. It obtains $CV_T$ by running **SD.Cover** $(\mathcal{BT}, RV)$.

  2. For each $S_{i,j} \in CV_T$, it sets $(GL_k, ML_k) = GMLabels(S_{i,j})$ and performs: It generates $SK_{IBE,S_{i,j}}$ by running **IBE.GenKey**$(GL_k \| T, MK_{IBE}, PP_{IBE})$. It receives $SK_{IBR,S_{i,j}}$ from $\mathcal{C}$ by submitting an identity $ML_k \| T$.

  3. It creates $UK_T = (CV_T, \{(SK_{IBE,S_{i,j}}, SK_{IBR,S_{i,j}})\}_{S_{i,j} \in CV_T})$ and gives $UK_T$ to $\mathcal{A}$.

- For a decryption key query with an identity $ID$ and time $T$, $\mathcal{B}$ proceeds as follows:

  1. It retrieves $SK_{ID} = SK_{HIBE}$ by querying a private key to its own oracle. It also retrieves $UK_T$ by querying an update key to its own oracle.

  2. Next, it generates a delegated key $DK_{HIBE}$ of $SK_{HIBE}$ by running **HIBE.DelegateKey** for $ID$ and $T$.

  3. It gives $DK_{ID,T} = (UK_T, DK_{HIBE})$ to $\mathcal{A}$.

- For a revocation query with an identity $ID$ and time $T$, $\mathcal{B}$ proceeds as follows: It adds $(ID, T)$ to $RL$ if $ID$ was not revoked before.

> **Challenge**: $\mathcal{A}$ submits a challenge identity $ID^*$, challenge time $T^*$, and two challenge messages $M_0^*, M_1^*$. $\mathcal{B}$ proceeds as follows:

1. It first select random $R_2$ and sets $R_{1,M_0} = M_0^* \oplus R_2, R_{1,M_1} = M_1^* \oplus R_2$.

2. Next, it generates $CT^*_{HIBE}$ by running **HIBE.Encrypt**$((ID^*, T^*), R_2, PP_{HIBE})$.

3. It obtains $PV_{ID^*}$ by running **SD.Assign** $(\mathcal{BT}, v_{ID^*})$ where a leaf node $v_{ID^*}$ is associated with $ID^*$. For each $S_{i,j} \in PV_{ID^*}$, it obtains $(GL_k, ML_k) = GMLabels(S_{i,j})$ and proceeds as follows:

- If $k < \rho$, then it selects random $R_{3,k}$ and sets $R_{4,k} = R_{1,M_1} \oplus R_{3,k}$, and then generates $CT^*_{IBE,S_{i,j}}$ by running **IBE.Encrypt**$(GL_k \| T^*, R_{3,k}, PP_{IBE})$ and $CT^*_{IBR,S_{i,j}}$ by running **IBR. Encrypt** $(ML_k \| T^*, R_{4,k}, PP_{IBR})$.

- If $k = \rho$, then it performs the follows:

    (a) It selects random $R_{3,k}$ and sets $R_{4,k,M_0} = R_{1,M_0} \oplus R_{3,k}, R_{4,k,M_1} = R_{1,M_1} \oplus R_{3,k}$.

    (b) It generates $CT^*_{IBE,S_{i,j}}$ by running **IBE.Encrypt**$(GL_k \| T^*, R_{3,k}, PP_{IBE})$.

    (c) It receives $CT^*_{IBR,S_{i,j}}$ from $\mathcal{C}$ by submitting a challenge identity $ML_k \| T^*$ and challenge messages $R_{4,k,M_0}, R_{4,k,M_1}$.

- If $k > \rho$, then it selects random $R_{3,k}$ and sets $R_{4,k} = R_{1,M_0} \oplus R_{3,k}$, and then generates $CT^*_{IBE,S_{i,j}}$ by running **IBE.Encrypt**$(GL_k \| T^*, R_{3,k}, PP_{IBE})$ and $CT^*_{IBR,S_{i,j}}$ by running **IBR.Encrypt** $(ML_k \| T^*, R_{4,k}, PP_{IBR})$.
It creates $CT^*_{PV} = \left(PV_{ID^*}, \left\{\left(CT^*_{IBE,S_{i,j}}, CT^*_{IBR,S_{i,j}}\right)\right\}_{S_{i,j} \in PV_{ID^*}}\right)$.

4. It gives a challenge ciphertext $CT^* = (CT^*_{PV}, CT^*_{HIBE})$ to $\mathcal{A}$.

**Phase 2**: Same as Phase 1.
**Guess**: Finally, $\mathcal{A}$ outputs a guess $\mu' \in \{0, 1\}$. $\mathcal{B}$ also outputs $\mu'$.

# 5 Instantiations

In this section, we show that our generic RIBE construction can be instantiated as real RIBE schemes by using bilinear maps or lattices.

## 5.1 RIBE from bilinear maps

Previously, many RIBE schemes using the CS method were directly constructed on bilinear maps [6, 7, 9]. In addition, an RIBE scheme using the SD method was also directly constructed on bilinear maps [15]. Recently, a generic construction for RIBE using the CS method was proposed by Ma and Lin [18]. Nonetheless, different generic construction for RIBE using the SD/LSD method is still an interesting method because it allows different RIBE instantiations by changing the underlying cryptographic schemes and allows RIBE schemes with shorter update keys. Here, we will look at different instantiations of RIBE using the SD/LSD method that provide selective security or adaptive security.

First, we instantiate an efficient RIBE scheme that provides selective security by following the generic construction. To do this, we choose the BB-IBE scheme of Boneh and Boyen [38] as the underlying IBE scheme, which provides selective security in the DBDH assumption. For underlying IBR scheme, we choose the efficient LSW-IBR scheme of Lewko et al. [26]. However, the IBR scheme for our generic construction only requires that the revoked set $R$ of ciphertexts just consists of a single revoked identity $ID$. Thus, we can derive a simplified LSW-IBR scheme which supports a single revoked identity, and this simplified IBR scheme provides selective security in the DBDH assumption [27]. Finally, we choose the two-level BB-HIBE scheme of Boneh and Boyen [38] that provides selective security in the DBDH assumption. The resulting RIBE scheme that uses the SD/LSD method provides selective security under the DBDH assumption.

We analyze the private key, update key, and ciphertext size of our generic RIBE scheme with the LSD method in an asymmetric bilinear group. In the MNT159 bilinear group, the size of the $\mathbb{G}$ group is 159 bits, and the size of the $\hat{\mathbb{G}}$ group and the $\mathbb{G}_T$ group is 954 bits. In the BB-IBE scheme, the private key size is $2|\mathbb{G}_2|$ and the ciphertext size is $2|\mathbb{G}| + |\mathbb{G}_T|$ where $|\mathbb{G}|$ denotes the size of a group element. In the LSW-IBR scheme, the private key size is $3|\hat{\mathbb{G}}|$ and the ciphertext size is $3|\mathbb{G}| + |\mathbb{G}_T|$. In the BB-HIBE scheme, the private key size is $2|\hat{\mathbb{G}}|$ and the ciphertext size is $3|\mathbb{G}| + |\mathbb{G}_T|$. In our RIBE scheme, the private key size is $2|\hat{\mathbb{G}}|$ since it consists of the private key of HIBE, and the update key size is $20 * r * |\hat{\mathbb{G}}|$ since it is composed of the IBE and IBR private keys associated with a cover set, and the ciphertext size is approximately $0.5 * \log^{1.5} N * (5|\mathbb{G}| + 2|\mathbb{G}_T|)$ since it consists of the IBE and IBR ciphertexts associated with a path set. Thus, if we set $N = 2^{32}$ and $r = 1000$, the private key size is 238 bytes, the update key size is 2385 kilobytes, and the ciphertext size is 30 kilobytes.

Next, we instantiate an RIBE scheme that provides adaptive security. To this security, we use the IBE scheme of Waters [30] which provides adaptive security under the DBDH and DLIN assumptions, the IBR of Okamoto and Takashima [39] which is derived from an NIPE scheme that provides adaptive security, and the two-level HIBE scheme of Waters [30] that provides adaptive security under the DBDH and DLIN assumptions. The resulting RIBE scheme provides adaptive security under these standard assumptions in prime-order bilinear groups. The previous adaptively secure RIBE scheme of Lee et al. [15] is built in composite-order bilinear groups, but this generic RIBE scheme is built in prime-order bilinear groups. However, this generic RIBE scheme has a small size of private keys and a large size of ciphertexts, whereas the RIBE scheme of Lee et al. has a small size of ciphertexts and a large size of private keys.

## 5.2 RIBE from lattices

A number of RIBE schemes in lattices have been previously proposed [8, 11, 14, 17]. Although the first lattice based RIBE scheme using the CS method did not provide decryption key exposure resistance (DKER), the new RIBE scheme using the CS method that allows DKER was recently proposed by using the delegation property of HIBE [8, 17]. In addition, a lattice based RIBE scheme using the SD method also has been proposed, but this scheme has a serious limitation such that the identity space is restricted to be small universe because the Lagrange interpolation technique is directly applied to lattices [11].

We use the previously proposed efficient lattice based IBE, IBR, and two-level HIBE schemes to instantiate a lattice based RIBE scheme using the SD method. For the underlying IBE and HIBE schemes, we choose efficient IBE and HIBE schemes of Agrawal et al. [40] that provide selective security in the LWE assumption. For the underlying simple IBR scheme, we choose the NIPE scheme of Katsumata and Yamada [41] which is derived from a linear functional encryption scheme. Note that an NIPE scheme is easily transformed into a simplified IBR scheme with a single revoked identity and this resulting IBR scheme provides selective security in the LWE assumption.

We compare our RIBE scheme with the SD method and the RIBE scheme directly designed by Cheng and Zhang [11]. Cheng and Zhang derived their RIBE scheme in lattices by applying the design principle of the RIBE scheme of Lee et al. [15] in bilinear maps. To use the technique of Lee et al., it is necessary to use the Lagrange interpolation to recover a polynomial value in decryption. In lattices, if Lagrange coefficients and noise values in ciphertexts are multiplied, then a large noise value is obtained in the decryption process, which should be removed to obtain a message. Since the resulting noise value is exponentially increased as the

size of the identity space increases, their RIBE scheme has a serious problem that only a small universe of identity can be accepted. Therefore, our RIBE scheme with the SD method is the first lattice based RIBE scheme using the SD method that supports a large universe of identity and provides the DKER property.

## 6 Conclusion

In this paper, we proposed a new generic RIBE construction with the SD method. Our generic construction uses an IBE scheme, an IBR scheme with single revoked identity, and a two-level HIBE scheme as building blocks. The generic RIBE construction can be instantiated by bilinear maps or lattices, and the private key consists of constant IBE and HIBE private keys, the update key consists of $O(r)$ number of IBE and IBR private keys, and the ciphertext mainly consists of $O(n^2)$ number of IBE and IBR ciphertexts. If our generic RIBE construction is extended to use the more efficient LSD method instead of the SD method, the ciphertext is reduced to $O(n^{1.5})$ number of IBE and IBR ciphertexts. In addition, if the underlying IBE, IBR, and HIBE schemes provide the CCA security and a one-time signature is used, then a CCA secure RIBE scheme can be generically constructed.

There are some interesting open problems. The first problem is to reduce the size of a ciphertext in our generic RIBE scheme with the SD method. In the previous generic RIBE scheme with the CS method, the size of a ciphertext can be reduced by using an IBBE scheme. In our generic RIBE scheme with the SD method, it is difficult to reduce the size of a ciphertext since it uses an IBR scheme. The second problem is to design a generic RHIBE scheme with the SD method. To design a generic RHIBE scheme, the private key delegation is needed. It is possible to extend the IBE scheme to support key delegation by using an HIBE scheme, but it is unclear how to extend the IBR scheme to support key delegation.

## Author Contributions

**Conceptualization:** Kwangsu Lee.

**Formal analysis:** Kwangsu Lee.

**Funding acquisition:** Kwangsu Lee.

**Methodology:** Kwangsu Lee.

**Project administration:** Kwangsu Lee.

**Software:** Kwangsu Lee.

**Validation:** Kwangsu Lee.

**Writing – original draft:** Kwangsu Lee.

**Writing – review & editing:** Kwangsu Lee.

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
