## [Decision Letter · Decision Letter 0]

2 Jun 2020

PONE-D-20-11744

A Generic Construction for Revocable Identity-Based Encryption with Subset Difference Methods

PLOS ONE

Dear Dr. Lee,

Thank you for submitting your manuscript to PLOS ONE. After careful consideration, we have decided that your manuscript does not meet our criteria for publication and must therefore be rejected.

I am sorry that we cannot be more positive on this occasion, but hope that you appreciate the reasons for this decision.

Yours sincerely,

He Debiao

Academic Editor

PLOS ONE

Reviewers' comments:

Reviewer's Responses to Questions

**Comments to the Author**

1. Is the manuscript technically sound, and do the data support the conclusions?

Reviewer #1: Partly

Reviewer #2: Partly

2. Has the statistical analysis been performed appropriately and rigorously? 

Reviewer #1: Yes

Reviewer #2: N/A

3. Have the authors made all data underlying the findings in their manuscript fully available?

Reviewer #1: No

Reviewer #2: Yes

4. Is the manuscript presented in an intelligible fashion and written in standard English?

Reviewer #1: Yes

Reviewer #2: Yes

5. Review Comments to the Author

Reviewer #1: In this work, the authors design a Revocable Identity-Based Encryption RIBE scheme by combining IBE, identity-based revocation (IBR), and two-level HIBE schemes. Overall, the work has technical depth and has potential.

- The solution is described in a pure mathematical and technical way, which makes it hard to grasp it for non-expert readers. I suggest adding a paragraph to explain the solution in plaintext in an accessible manner.

- It would be quite interesting to compare the proposed solution with regards to other security solutions such as trust. In this context, I suggest comparing with the following articles:

• Towards trustworthy multi-cloud services communities: A trust-based hedonic coalitional game. IEEE Transactions on Services Computing, 11(1), 184-201, 2016.

• A survey on trust and reputation models for Web services: Single, composite, and communities. Decision Support Systems, 74, 121-134, 2015.

• Trust management for secure cognitive radio vehicular ad hoc networks. Ad Hoc Networks, 86, 154-165, 2019.

• A lightweight trust management algorithm based on subjective logic for interconnected cloud computing environments. The Journal of Supercomputing, 75(7), 3534-3554, 2019.

- The authors should present some experimental results to validate the performance of the solution.

Reviewer #2: Based on my own reading of manuscript no.PONE-D-20-11744,

I think that there is not enough novelty in the paper, most of the contribution has been published in the following paper:

Kwangsu Lee:

A Generic Construction for Revocable Identity-Based Encryption with Subset Difference Methods. IACR Cryptol. ePrint Arch. 2019: 798 (2019)

https://eprint.iacr.org/2019/798

6. PLOS authors have the option to publish the peer review history of their article (what does this mean?). If published, this will include your full peer review and any attached files.

Reviewer #1: No

Reviewer #2: No

- - - - -

---

## [Decision Letter · Decision Letter 1]

22 Jul 2020

PONE-D-20-11744R1

A Generic Construction for Revocable Identity-Based Encryption with Subset Difference Methods

PLOS ONE

Dear Dr. Lee,

Thank you for submitting your manuscript to PLOS ONE. After careful consideration, we feel that it has merit but does not fully meet PLOS ONE’s publication criteria as it currently stands. Therefore, we invite you to submit a revised version of the manuscript that addresses the points raised during the review process.

Please address the issues raised by Academic editor Dr. Hua Wang and the reviewers.

We look forward to receiving your revised manuscript.

Kind regards,

Qi Jiang

Academic Editor

PLOS ONE

Journal Requirements:

Kwangsu Lee was supported as part of Military Crypto Research Center

(UD170109ED) funded by Defense Acquisition Program Administration (DAPA) and

Agency for Defense Development (ADD). NO

i) Please provide an amended statement that declares *all* the funding or sources of support (whether external or internal to your organization) received during this study, as detailed online in our guide for authors at http://journals.plos.org/plosone/s/submit-now.

ii) Please also include the statement “There was no additional external funding received for this study.” in your updated Funding Statement.

iii) Please include your amended Funding Statement within your cover letter. We will change the online submission form on your behalf.

3. Thank you for updating your data availability statement. You note that your data are available within the Supporting Information files, but no such files have been included with your submission. At this time we ask that you please upload your minimal data set as a Supporting Information file, or to a public repository such as Figshare or Dryad.

Please also ensure that when you upload your file you include separate captions for your supplementary files at the end of your manuscript.

As soon as you confirm the location of the data underlying your findings, we will be able to proceed with the review of your submission.

Additional Editor Comments (if provided):

Comment from Dr. Hua Wang：

The paper is based on the work published by Inscrypt 2019 which was also a good research work. I can see the contributions of the paper but the motivations of the paper are not explicitly described. One paragraph of motivations is necessary to show the importance of the research work and also attract more readers in future.

Reviewers' comments:

Reviewer's Responses to Questions

**Comments to the Author**

1. If the authors have adequately addressed your comments raised in a previous round of review and you feel that this manuscript is now acceptable for publication, you may indicate that here to bypass the “Comments to the Author” section, enter your conflict of interest statement in the “Confidential to Editor” section, and submit your "Accept" recommendation.

Reviewer #3: All comments have been addressed

Reviewer #4: All comments have been addressed

2. Is the manuscript technically sound, and do the data support the conclusions?

Reviewer #3: Yes

Reviewer #4: Partly

3. Has the statistical analysis been performed appropriately and rigorously? 

Reviewer #3: Yes

Reviewer #4: N/A

4. Have the authors made all data underlying the findings in their manuscript fully available?

Reviewer #3: Yes

Reviewer #4: Yes

5. Is the manuscript presented in an intelligible fashion and written in standard English?

Reviewer #3: Yes

Reviewer #4: Yes

6. Review Comments to the Author

Reviewer #3: The author has addressed the questions in the previous version.

But there are still some minor isisues needed to be solved:

1. I suggest that the section 1.2 should be demonstrated before the concrete construction.

2. Compared with the reference 15, what are the advantages of proposed generic RIBE scheme?

Reviewer #4: Paper strengths.

This manuscript mainly focuses on the demand of key revocation for identity-based encryption, and proposes a revocable solution with the subset difference method. Their IBE scheme combine identity-based revocation (IBR), and two-level HIBE schemes as basic building blocks.

Strength:

1) The motivation and idea are good, and the proposed RIBE approach seems to be innovative and practical.

2) The security definition and formal proof of designed RIBE is very nice.

3) The manuscript is presented in an intelligible fashion and is written in standard English.

Paper weaknesses.

There is no implementation or complexity analysis to indicate the feasibility of the scheme.

Detailed review.

1). In the Definition 2.5 (Hierarchical Identity-Based Encryption, HIBE), what's the relation between ID|\\ell, ID|k and ID'_k? Dose the ID|\\ell denote a subset of ID|k or the opposite? Is the ID'_k the last identity of hierarchical identity ID|k or some identity of ID|k ?

2). In RIBE.Setup (Section 3.3), “It defines a binary tree BT by running SD.Setup(2n) where an identity ID \\in I is uniquely assigned to a leaf node v such that Label(v) = ID.” What does the identity “ID” point to? Do the authors mean all the identities that users use are pre-set by the key generation center?

3). The authors mention that “the size of an update key is reduced but the size of a ciphertext is increased.”. It is an interesting improvement. I suggest the authors could provide concrete analysis on computation and storage complexity or even some experimental implementation in the Section 5. This will make readers fully understand this paper's innovation.

7. PLOS authors have the option to publish the peer review history of their article (what does this mean?). If published, this will include your full peer review and any attached files.

Reviewer #3: No

Reviewer #4: No

---

## [Decision Letter · Decision Letter 2]

31 Aug 2020

A Generic Construction for Revocable Identity-Based Encryption with Subset Difference Methods

PONE-D-20-11744R2

Dear Dr. Lee,

We’re pleased to inform you that your manuscript has been judged scientifically suitable for publication and will be formally accepted for publication once it meets all outstanding technical requirements.

Kind regards,

Qi Jiang

Academic Editor

PLOS ONE

Additional Editor Comments (optional):

Reviewers' comments:

Reviewer's Responses to Questions

**Comments to the Author**

1. If the authors have adequately addressed your comments raised in a previous round of review and you feel that this manuscript is now acceptable for publication, you may indicate that here to bypass the “Comments to the Author” section, enter your conflict of interest statement in the “Confidential to Editor” section, and submit your "Accept" recommendation.

Reviewer #3: All comments have been addressed

Reviewer #4: (No Response)

2. Is the manuscript technically sound, and do the data support the conclusions?

Reviewer #3: Yes

Reviewer #4: (No Response)

3. Has the statistical analysis been performed appropriately and rigorously? 

Reviewer #3: Yes

Reviewer #4: (No Response)

4. Have the authors made all data underlying the findings in their manuscript fully available?

Reviewer #3: Yes

Reviewer #4: (No Response)

5. Is the manuscript presented in an intelligible fashion and written in standard English?

Reviewer #3: Yes

Reviewer #4: (No Response)

6. Review Comments to the Author

Reviewer #3: The authors have addressed my questions in previous version, then this paper at current version can be accepted.

Reviewer #4: (No Response)

7. PLOS authors have the option to publish the peer review history of their article (what does this mean?). If published, this will include your full peer review and any attached files.

Reviewer #3: No

Reviewer #4: No

---

## [Editor Report · Acceptance letter]

3 Sep 2020

PONE-D-20-11744R2 

A Generic Construction for Revocable Identity-Based Encryption with Subset Difference Methods  

Dear Dr. Lee:

I'm pleased to inform you that your manuscript has been deemed suitable for publication in PLOS ONE. Congratulations! Your manuscript is now with our production department. 

Kind regards, 

on behalf of

Dr. Qi Jiang 

Academic Editor

PLOS ONE